

# Trans-Pacific transport and evolution of aerosols: Spatiotemporal characteristics and source contributions

Zhiyuan Hu[1], Jianping Huang[1], Chun Zhao[2], Yuanyuan Ma[3], Qinjian Jin[4], Yun Qian[5], L. Ruby Leung[5], Jianrong Bi[1], Jianmin Ma[1]

[1]Key Laboratory for Semi-Arid Climate Change of the Ministry of Education, College of Atmospheric Sciences, Lanzhou University, Lanzhou 730000, China
[2]School of Earth and Space Sciences, University of Science and Technology of China, Hefei, Anhui, China.
[3]Key Laboratory of Land-surface Process and Climate Change in Cold and Arid Regions, Northwest Institute of Eco-environment and Resources, Chinese Academy of Science, Lanzhou 730000, China
[4]Department of Earth and Atmospheric Sciences, Cornell University, Ithaca, New York 14853, USA
[5]Atmospheric Sciences and Global Change Division, Pacific Northwest National Laboratory, Richland, WA, USA

*Correspondence to*: Jianping Huang (hjp@lzu.edu.cn)

**Abstract.** Aerosols in the mid- and upper-troposphere have a long enough lifetime for trans-Pacific transport from East Asia

to North America to influence air quality in the West Coast of the United States (US). Here, we conduct quasi-global simulations (180° W – 180° E and 70° S – 75° N) from 2010 to 2014 using an updated version of WRF-Chem (Weather Research and Forecasting model fully coupled with chemistry) to analyze the spatiotemporal characteristics and source contributions of trans-Pacific aerosol transport. We find that trans-Pacific total aerosols have a maximum mass concentration (about 15 $\mu$g m$^{-3}$) in the boreal spring with a peak between 3 and 4 km above the surface around 40° N. Sea-salt and dust

dominate the total aerosol mass concentration below 1 km and above 4 km, respectively. About 80.8 Tg of total aerosols (48.7 Tg of dust) are exported annually from East Asia, of which 26.7 Tg of aerosols (13.4 Tg of dust) reach the West Coast of the US. Dust contributions from four desert regions in the Northern Hemisphere are analyzed using a tracer-tagging technique. About 4.9, 3.9, and 4.5 Tg year$^{-1}$ of dust aerosol emitted from North Africa, Middle East and Central Asia, and East Asia, respectively, can be transported to the West Coast of the US. The trans-Pacific aerosols dominate the column-

integrated aerosol mass (~65.5%) and number concentration (~80%) over the western North America. Radiation budget analysis shows that the inflow aerosols could contribute about 86.4% (–2.91 W m$^{-2}$) at the surface, 85.5% (+1.36 W m$^{-2}$) in the atmosphere and 87.1% (–1.55 W m$^{-2}$) at the top of atmosphere to total aerosol radiative effect over western North America. However, near the surface in the central and eastern North America, aerosols are mainly derived from local emissions and the radiative effect of imported aerosols decreases rapidly. This study motivates further investigations of the

potential impacts of trans-Pacific aerosols from East Asia on regional air quality and hydrological cycle in North America.



## 1 Introduction

Atmospheric aerosols, the liquid or solid particulate matter in the atmosphere, are known to be a crucial forcing agent of weather and climate at regional and global scales (Lau et al., 2008; Jimenez et al., 2009; Bond et al., 2013; Huang et al., 2014; Zhao et al., 2011, 2014). Aerosols can change the earth's energy budget directly by absorbing and scattering solar

radiation (Balkanski et al., 2007; Zhao et al., 2010; Jin et al., 2014, 2015; Bi et al., 2017) and indirectly by acting as cloud condensation nuclei (CCN) or ice nuclei (IN) and influence the properties of the cloud (Huang et al., 2006; Creamean et al., 2013; Jin et al., 2016, 2018; Li et al., 2018). When emitted from North Africa, Europe, and East Asia, aerosols could be lifted into the mid- and upper-troposphere and subsequently transported by the strong westerlies over the North Pacific basin to North America (Yienger et al., 2000; Holzer et al., 2003, 2005; Liang et al., 2004; Wuebbles et al., 2007; Hu et al., 2016).

The trans-Pacific aerosols can affect atmospheric composition (Chin et al., 2007; Yu et al., 2008), stratospheric ozone depletion (Solomon, 1999), surface air quality (VanCuren, 2003; Heald et al., 2006; Chin et al., 2007; Yu et al., 2012; Tao et al., 2016), regional visibility (Watson, 2002), human health (Pope, 2000; Pope et al., 2002; Schwartz, 1994), regional climate (Eguchi et al., 2009; Huang et al., 2009, 2011; Yu et al., 2012; Fan et al., 2014, 2015), and ecological integrity (Bytnerowicz et al., 1996; Schindler, 1988, 1999) in downwind regions, such as the United States (US).

Trans-Pacific aerosols are complex mixtures of natural and anthropogenic aerosols and may potentially impact the western US in many ways (Jaffe et al., 1999; Jacob et al., 2003; Huebert et al., 2003; Parrish et al., 2004; ; Chin et al., 2007; Fairlie et al., 2007, 2009; Fischer et al., 2009; Singh et al., 2009; Yu et al., 2008, 2012; Hu et al., 2016). Therefore, a number of observation campaigns (e.g., Jacob et al., 2003; Eguchi et al., 2009; Huang et al., 2008; Uno et al., 2011; Yu et al., 2008, 2012) and modeling projects (e.g., Park et al., 2005; Heald et al., 2006; Chin et al., 2007; Hadley et al., 2007; Alizadeh-

Choobari et al., 2014; Hu et al., 2016) were undertaken to understand the characteristics and impacts of trans-Pacific aerosols. Previous studies found that aerosols could traverse the Pacific Ocean in about 7–10 days (Eguchi et al., 2009) with the largest efficiency in spring (Takemura et al., 2002; Huang et al., 2008; Yu et al., 2012; Eguchi et al., 2009; Uno et al., 2009, 2011). As a major composition of aerosols, mineral dust plays an important role during the trans-Pacific transport. Eguchi et al. (2009) revealed that dust from the Gobi and Taklimakan deserts contributed significantly to the trans-Pacific dust amount,

with the Taklimakan dust transported at higher altitudes than Gobi dust. At the surface, trans-Pacific dust increases the fine particle concentration by 5−24% over the western US on annual mean, which is about 3 to 4 times more than the transport of anthropogenic pollution to the US on annual average (Chin et al., 2007). Yu et al. (2012) used MODIS-CALIOP to estimate the trans-Pacific dust fluxes and shown that about 56 Tg of East Asian dust could reach the western US every year. Anthropogenic aerosols also have an impact comparable to dust during trans-Pacific transport (Takemura et al., 2002;

Hadley et al., 2007). It has also been found that trans-Pacific pollutants could increase aerosol concentrations by about 0.2 $\mu$g m$^{-3}$ (Chin et al., 2007) over the western US, adding about 0.16 $\mu$g m$^{-3}$ of sulfate over northwestern US (Heald et al., 2006) and increasing black carbon (BC) amount by more than 70% of locally-emitted BC in North America (Hadley et al., 2007).



Trans-Pacific aerosols have significant impact on the climate system and surface air quality through absorbing and scattering of terrestrial and solar radiation (Yu et al., 2012), and modifying cloud and precipitation processes (Ault et al., 2013; Creamean et al., 2013) over western US. For example, the imported aerosols reduce cloud-free net solar radiation by –1.7 Wm$^{-2}$ at top of atmosphere (TOA) and –3.0 Wm$^{-2}$ at the surface (SFC) (Yu et al., 2012). The imported pollutants

account for 31% to 59% of the direct radiative forcing induced by imported dust, because they are more effective in absorbing and scattering solar radiation (Yu et al., 2012). Ault et al. (2011) showed that trans-Pacific dust could increase precipitation by serving as effective ice nuclei and modifying the high-altitude precipitating orographic clouds during the CalWater field campaign. Creamean et al. (2013) demonstrated that the trans-Pacific transported biological aerosols from Saharan and Asian deserts also played an important role in orographic precipitation processes as ice nuclei in the western US.

Further, the trans-Pacific transport aerosols can change the stability of the atmospheric boundary layer by absorbing solar radiation (Ramanathan et al., 2008) and accelerate snowmelt and influence the regional climate and hydrological cycle through deposition on snowpack (Painter et al., 2010; Qian et al., 2009, 2015) in western US mountains.

Previous studies have provided a growing sense of the impact of trans-Pacific transport aerosols over the West Coast of the US (Jaffe et al., 1999; Jacob et al., 2003; Chin et al., 2007; Yu et al., 2002, 2008, 2012). Most cases reported that the

composition and spatial distribution of aerosols were significantly determined by aerosol sizes and number concentration, which could affect cloud formation and distribution (e.g. droplet size, water phase). Serving as effective cloud condensation nuclei, aerosols can potentially enhance or weaken precipitation over the western US (Rosenfeld et al., 2001; Fan et al., 2004; Kelly et al., 2007; Koehler et al., 2007; Eguchi et al., 2009; Creamean et al., 2013). Changing aerosol particle sizes can change the radiative budget through absorbing and scattering solar radiation (Liao and Seinfeld, 2005; Kim et al., 2004), and

increasing aerosol number concentration can affect surface air quality (Hu et al., 2016). However, the detailed composition and spatiotemporal characteristics of trans-Pacific aerosols over the West Coast of the US, which are critical for investigating aerosol impact, are not well understood. Also, aerosol sizes and number concentrations are not well studied because few analyses are performed on the individual aerosol composition (e.g. dust or sulfate) or the total aerosol quantities. Aerosol aging after long range transport can change the aerosol compositions and distribution, and increase the uncertainty

in estimating aerosol radiative forcing (RF) by more than 100% or even 200%, which is more significant in regional scale than in global scale (Kulmala et al., 2009). More importantly, we quantify the various source contributions in regional climate model, which has better capabilities on simulating the large geographical variability and aerosol–cloud–precipitation interaction (Zhao et al., 2013a; Hu et al., 2016). Given the significant amount of dust aerosols from East Asia, Middle East and Central Asia, and North Africa, a better understanding of the source-receptor relationship of dust and the dust source

contributions from various desert regions in the Northern Hemisphere to the aerosol abundance in West Coast of the US are warranted.

In this study, we use an updated version of Weather Research and Forecasting (WRF) model with chemistry (WRF-Chem; Grell et al., 2005) developed at the University of Science and Technology of China (USTC) for quasi-global simulation as described by Hu et al. (2016, 2019). The quasi-global simulation of trans-Pacific transport of aerosols with the USTC



version of WRF-Chem has been evaluated by Hu et al. (2016). In this paper, we analyze the experiments with a focus on (1) the characteristics of trans-Pacific aerosols including the spatiotemporal distributions of their chemical compositions; (2) the relative contributions to aerosols from the local emissions in the western US and the long-range transport from East Asia; (3) the contributions from major deserts (i.e., North America, East Asia, North Africa and the elsewhere in the world (mainly for

Middle East and Central Asia)) to the dust loading over the western US. Section 2 describes the methodology. Section 3 presents the results of the spatiotemporal characteristics of trans-Pacific aerosols and their impacts on aerosol properties over the west US. The conclusions and discussion are provided in Section 4.

## 2 Methodology

### 2.1 WRF-Chem

The version of WRF-Chem from USTC is used in this study. The MOSAIC aerosol scheme (Model for Simulation Aerosol Interactions and Chemistry) (Zaveri et al., 2008) is configured in our simulations, which has been coupled with the CBM-Z (i.e., carbon bond version Z) photochemical mechanism (Zaveri and Peters, 1999). MOSAIC uses a bin approach with eight discrete size bins to represent aerosol size distributions (Fast et al., 2006). There are eight kinds of aerosol compositions simulated by the model, including mineral dust, nitrate ($NO_3^-$), sulfate ($SO_4^{-2}$), organic matter (OM), black carbon (BC),

ammonium ($NH_4^+$), sea-salt, and water mixed with other aerosol compositions. The physical and chemical processes of aerosols are also included in MOSAIC aerosol scheme. The approach of aerosol dry deposition is followed Binkowski and Shankar (1995). Wet removal of aerosols is simulated by the approach of Easter et al. (2004) and Chapman et al. (2009). Moreover, aerosol optical properties (e.g., asymmetry factor for scattering, extinction, single scattering albedo (SSA)) of each model grid box are computed by a function of wavelength.

### 2.2 Numerical experiments

The simulations are performed at 1° horizontal resolution with $360 \times 145$ grid cells (180° W – 180° E, 67.5° S – 77.5° N) for the period of 2010–2014, and they are configured with 35 vertical layers from the surface up to 50 hPa. The lateral meridional boundary and meteorological initial conditions are derived from the National Center for Environmental Prediction final analysis (NCEP/FNL) data at 1° horizontal resolution and 6-hr temporal intervals. Dust emission follows

Ginoux et al. (2001), which is based on the GOCART dust emission scheme, and the sea-salt emission is calculated as Zhao et al. (2013a) based on sea surface temperature (Jaeglé et al., 2011) and the correction of particles with radius less than 0.2 $\mu$m (Gong, 2003). Anthropogenic emissions are obtained from the Hemispheric Transport of Air Pollution version-2 (HTAPv2) with monthly temporal resolution and the $0.1° \times 0.1°$ horizontal resolution for each year of 2010–2014 (Janssens-Maenhout et al., 2015). Biomass burning emissions are obtained from the Fire INventory from NCAR (FINN) at 1 km × 1

km horizontal resolution and one hourly temporal resolution (Wiedinmyer et al. 2011). Also, Morrison 2-moment microphysics scheme, RRTMG longwave and shortwave radiation schemes, Morrison 2-moment microphysics scheme,



CLMv4.0 (Community Land Model) land surface scheme, and the MYJ (Mellor-Yamada-Janjic) planetary boundary layer scheme are used in this study.

To understand the impact of aerosol trans-Pacific transport on local aerosol properties over the western US, this study analyzes two sets of simulations. The standard simulation includes all natural and anthropogenic emissions across the world, which has been described in details by Hu et al. (2016). The sensitivity simulation excludes all the emissions from North America (10° N – 70° N and 170° W – 60° W). The difference between the two simulations represents the impact of local emissions over North America. In addition, because the trans-Pacific transported dust has an important impact on air quality and weather over the western US (e.g., Fischer et al., 2009; Creamean et al., 2013), this study applies a tracer-tagging method in the WRF-Chem simulations to quantify the dust contributions during the trans-Pacific transport. This method tags and explicitly tracks dust particles emitted from the independent regions of major deserts within a single simulation through using additional model variables. Four dust source regions are selected in the model, i.e., North America (NAM, 15° N – 50° N and 80° W – 140° W), East Asia (EAS, 25° N – 50° N and 75° E – 150° E), North Africa (NAF, 0° N – 40° N and 20° W – 35° E), and the elsewhere in the world (EIW). It is noted that the advective and physical tendencies of tagged dust variables are simulated in the same way as dust aerosols.

# 3 Results and discussion

## 3.1 Characteristics of trans-Pacific aerosols

### 3.1.1 Spatial distribution of total mass

Firstly, Hu et al. (2016) evaluated the simulations with various satellite retrievals and surface measurements, including AOD from MODIS, MISR, OMI and AERONET, aerosol extinction coefficients from CALIPSO, and the surface mass concentration from IMPROVE. These results shown that the simulations well captured the spatial distribution and vertical profile characteristics of trans-Pacific transport aerosols. Figure 1 shows the spatial distribution of seasonal mean aerosol mass concentrations across the Pacific Ocean from the simulation averaged for the period of 2010–2014. The trans-Pacific transport of aerosol mass concentrations has similar spatial distribution and seasonal variability as that of AOD discussed by previous studies (e.g., Chin et al., 2007; Yu et al., 2008, 2012; Hu et al., 2016). Aerosols export from the East Asian continent, a gradual decrease of aerosol concentrations across the Pacific Ocean, and import of aerosols into North America are clearly seen in all seasons. In Taklimakan and Gobi (for dust) and India and South Asia (for anthropogenic aerosol), significant amounts of aerosols are produced, and the aerosol mass loadings can reach 320 mg m$^{-2}$ (Fig. 1). During long-range transport, aerosols with high mass concentrations coincide with the sub-tropical jet and reach the west coast of the US. The peak trans-Pacific aerosol mass concentrations occur in spring (MAM) due to the offset by the westward transport (particularly south of 30° N), while the minimum occurs in summer (JJA) because of the greatest aerosol removal induced by summer monsoon circulation and precipitation (Holzer et al., 2005; Yu et al., 2008, 2013). The modeling results indicate that



the spatial distribution of aerosols is similar in all seasons. About 25% of exported aerosols from Asia arrive the West Coast of the US (Fig. 1), which is consistent with the MODIS-CALIOP assessment (Yu et al., 2008). In general, the strong spatial and seasonal variations of aerosol mass concentrations are due to the seasonal variations of emissions of aerosols and their precursors, the mid-latitude westerlies, and extratropical cyclones (Yu et al., 2008, 2012; Fast et al., 2014; Hu et al., 2016).

The seasonal variation of vertical cross-section of zonal-mean aerosol mass concentrations averaged for 2010–2014 shows large latitudinal and vertical gradients in aerosol mass concentrations appear through all seasons (Fig. 2). In MAM, the maximum aerosol mass concentration occurs below 4 km with about 15 $\mu$g m$^{-3}$ within the 27° N – 44° N segment, consistent with previous research results (e.g., Huang et al., 2014; Hu et al., 2016). From 1 km to 8 km, aerosol mass concentrations are about 10 $\mu$g m$^{-3}$ in the 35° N – 45° N segment, which is the largest compared to the other three seasons. In winter (DJF),
there are about 4 $\mu$g m$^{-3}$ of aerosol mass concentrations below 2-6 km across 25° N to 50° N. For JJA and SON, the segment of higher concentrations shifts further north to 35° N – 50° N, potentially related to the northward shift of the easterlies and the reduced strength of the westerlies, along with precipitation removal increasing in the outflow region (Yu et al., 2008). Similar to the horizontal spatial distribution (Fig. 1), the vertical distribution of aerosol mass concentrations is the largest in MAM due to the strong frontal and postfrontal convection and atmospheric boundary layer turbulent mixing that lift more
aerosols to the troposphere (Yu et al., 2008). Note that the largest aerosol mass concentration appear below 1 km in the seasons that is dominated by seasalt (figure not shown).

**3.1.2 Spatial distribution of aerosol components**

Figure 3 shows the annual mean spatial distribution of column integrated mass of various aerosol compositions. The amounts of trans-Pacific aerosols are estimated from the sensitivity simulation with all emissions from North America
excluded. The results show that the trans-Pacific aerosols can reach the western US, and even the central US. Four types of dust tracers emitted from the four regions of NAM, EAS, NAF, and EIW with major deserts (defined in Section 2) are also shown. With the dust tracking method, the dust from multiple sources mixed in the outflow of Asia (Chin et al., 2007) can be isolated. The result shows that dust from EAS, NAF, and EIW are the major contributors to the trans-Pacific dust over the North Pacific. The NAF dust is not only transported across northern Africa into Europe (Park et al., 2005; Lee et al., 2010),
but also has been found to mix with EIW dust (mainly from the Middle East and Central Asian dust sources) and could be carried eastward to East Asia (Hu et al., 2018) (Fig. 3). Over East Asia, the EAS dust is mainly from local sources (e.g., Taklimakan and Gobi desert), followed by the NAF dust and EIW dust from inter-continental transport (e.g., Sahara and Arabian desert). Among the three major tagged source regions, the EAS dust (22.34 mg m$^{-2}$) contributes about 28.3% to the total aerosol mass loading over East Asia. The NAF dust (8.71 mg m$^{-2}$) contributes about 11.0%, and the EIW dust (7.11 mg
m$^{-2}$) contributes about 9.0%, respectively. Other aerosol compositions (e.g., sulfate, nitrate, ammonium, organic matter, black carbon, other inorganic matter and sea-salt) also have significant contributions to the trans-Pacific transport aerosols with about 51.6% contribution over East Asia.



The EAS dust decreases rapidly (7.60 mg m$^{-2}$) during transport because of dry deposition (gravitational sedimentation and turbulent mix-out) near the source region (Zhao et al., 2012; Huang et al., 2014) and wet deposition (precipitation scavenging) (Easter et al., 2004; Chapman et al., 2009) during the long-range transport. For NAF dust (6.24 mg m$^{-2}$) and EIW dust (5.07 mg m$^{-2}$), they reside mainly in the upper troposphere, which results in less dry and/or wet deposition, so they

contribute about 13.7% and 11.1% to total aerosol mass loading. The mass loading of other aerosol compositions is 26.67 mg m$^{-2}$, which is about 58.6% of total aerosol mass concentration.

In the West Coast of North America, dust contributes about 41.1% compared to other aerosol mass loading, followed by sea-salt of about 21.7%. For the dust contribution, EAS dust (4.79 mg m$^{-2}$), NAF dust (4.59 mg m$^{-2}$) and EIW dust (3.60 mg m$^{-2}$) contribute 14.6%, 14.0% and 11.0% to total aerosol mass loading, respectively. However, the mass loading of NAM

dust (0.48 mg m$^{-2}$) is far lower than the trans-Pacific dust. Also, the other aerosol compositions are 19.24 mg m$^{-2}$. In general, the trans-Pacific transport dust is about 1~2 times larger of the aerosol mass loading of pollution transport to this region, of which the transport dust at the surface is 2~4 times higher than that of pollution as discussed by Chin et al. (2007).

Figure 4a-c illustrates the cross-section of zonal aerosol mass concentration vertical distribution from the WRF-Chem model simulation over three sub-regions (the western Pacific: 20° N – 50° N and 120° E – 140° E; the central Pacific: 20° N

– 50° N and 140° E – 140° W; the eastern Pacific: 20° N – 50° N and 140° W – 120° W) (shown as the black boxes in Fig. 1). Aerosol mass concentrations decrease evidently with elevation along the Pacific Ocean with dramatic reduction from around 15 to 1 $\mu$ g m$^{-3}$ from the surface to about 10 km. Over the western Pacific, the aerosol mass concentration is the highest below 2 km in the 30° N – 45° N segment with a peak value of 15 $\mu$g m$^{-3}$. The higher aerosol mass concentrations are attributed to outflow of dust and pollution aerosols in this region (e.g., Yu et al., 2012; Huang et al., 2014, Hu et al.,

2016). Over the central and eastern Pacific, because of wet/dry deposition (Zhao et al., 2012), the aerosol mass concentrations decrease rapidly from that of the western Pacific. Larger aerosol mass concentrations occur below about 1 km in the central Pacific (about 10 $\mu$g m$^{-3}$) and about 0.8 km in the eastern Pacific (about 8 $\mu$g m$^{-3}$). To characterize the aerosol vertical distribution, we show the profiles of total aerosol mass concentrations and vertical distributions of aerosol composition fractions in Fig.4 d-f. Over the Pacific Ocean, Sea-salt is the dominant aerosol below 1 km, while over the

western Pacific, the dust and pollution aerosols outflow from East Asia make significant contributions. Above 4 km, dust dominates the aerosol mass concentrations in all three sub-regions. Significantly, the EAS dust is the dominant dust contributor below 4 km, and the NAF dust and EIW dust are comparable to the EAS dust above 4 km. During the trans-Pacific transport, the NAF dust and EIW dust increase with altitude and contribute aerosol mass concentrations that is close to the EAS dust above 2 km. Overall, the EAS dust is the major dust aerosol over the eastern and central Pacific below 1 km

(exclude sea-salt). Above 2 km, dust from all three sources has similar contribution. Notably, anthropogenic aerosols (i.e., sulfate and nitrite) also make significant contributions over the three sub-regions, even above 2 km. Over the western Pacific, the anthropogenic aerosols contribute about more than 40% mass concentrations above 2 km, in which the largest is sulfate with 15% contribution and followed by OIN (12%) and OC (8%). Over the central and eastern Pacific, the anthropogenic aerosols contribution is similar above 2 km, in which the sulfate is about 15%, followed by OIN (8%) and OC (6%).



### 3.1.3 Mass versus number concentrations

The size of aerosol particles can range from nanometer to micrometer, which is a critical factor influencing clouds, precipitation and surface air quality (Yu et al., 2008; Zhao et al., 2013b). However, few studies have focused on aerosol number loading during the trans-Pacific transport. Zhao et al. (2013b) demonstrated that the WRF-Chem model configured

with 8 size bins could appropriately describe the aerosol size distributions and the changes of mass fraction of coarse and fine particles during the long-range transport. Figure 5 shows the fractional contributions of different aerosol compositions to seasonal and annual mean aerosol mass and number over the three sub-regions. Aerosol mass and number concentrations show significantly different fractions even for the same aerosol species in various regions. They decrease along the Pacific Ocean from west to east, with a peak in MAM. Dust dominates the total aerosol mass concentrations and account for more

than 50% in MAM (Fig. 5). Over the western Pacific, the EAS dust is the dominant dust aerosol, but for the central and eastern Pacific, dust from different sources has more comparable contributions. In DJF, the NAF dust and EIW dust mass is greater than that of the EAS dust over the eastern Pacific, but they have similar mass in JJA and SON. Hence it is shown clearly that the dust outflow from EAS is not originated from the EAS local region alone, but also from NAF and EIW (Middle East and Central Asia).

Unlike the aerosol mass contributions, the aerosol number contributions are dominated by fine sulfate particles. Over the western Pacific, the contribution of sulfate is more than 45%, with a peak in JJA at 50%. This is attributed to the higher photochemical activity in the warm season (Hu et al, 2016). During the transport over the central and eastern Pacific, the contribution of sulfate becomes more than 60%, but the nitrate decreases rapidly to less than 4%, which is likely due to the fact that nitrate is mainly concentrated in the low level over the western Pacific and can be removed easily during the

transport. The aerosol number concentrations are dominated by fine sulfate particles, followed by fine ammonium and organic matter particles. Compared with fine pollution aerosols, the dust aerosol number contribution is much less (less than 1%) due to the much larger particle size. Furthermore, Chin et al. (2007) found that sulfate from Europe is the main source of the trans-Pacific transport pollution aerosol, and more sulfate particles remain at the higher altitude when they are imported to the eastern Pacific. Therefore, only limited sulfate could be removed during the transport and most of the sulfate

particles can reach the eastern Pacific.

Figure 6 shows the vertical distribution of mass and number fractions of various aerosol compositions. Dust mass from three sources dominates the total aerosol mass concentrations above 4 km (about 60%). Sea-salt is the dominant aerosol species that contributes more than 40% of mass over the central and eastern Pacific below 2 km. Also, the EAS dust mass is distributed throughout the column with a peak at 2 km, but the NAF dust and EIW dust are distributed mainly above 4 km

over the western Pacific. Over the eastern Pacific, the EAS dust and NAF dust mass is mainly located above 1 km. For particle number, sulfate provides larger contribution in the column atmosphere and organic matter is similar to sulfate over the western Pacific, which is attributed to biomass burning and the use of coal in Asia (Bian et al., 2007; Yu et al., 2008).



During the transport, the aerosol number changes very little because of the minimal wet removal above 400 hPa, but there are larger changes because of precipitation removal below 800 hPa (not shown).

To better understand the vertical profiles of aerosol size distribution, the modeled size distributions of aerosol mass and number over the three sub-regions averaged for 2010–2014 are shown in Figure 7. Aerosols of size 2.5~10.0 μm contribute mainly to aerosol mass below 1 km, with a maximum at 5.0 μm (about 40%). Above 2 km, aerosols of size 1.25~5.0 μm (Bin4-Bin7) become the major size range with more than 35% contribution. In the range of 0.312~1.25 μm, the contribution is about 15%. There is a significant aerosol mass distribution in Bin4 (0.312~0.625 μm) above the surface to 4 km over the western Pacific. For aerosol number, the 0.039~0.156 μm (Bin1-Bin3) aerosols are the main contributors and dominate the aerosol number during aerosol trans-Pacific transport. The Bin1 fraction increases with altitude over the western Pacific, which indicates clearly the decreasing aerosol size with increasing altitude. However, during the transport pathway, aerosol size changes little over the central and eastern Pacific which are dominated by fine particles (0.039~0.156 μm). This analysis suggests that the aerosol mass is dominated by Bin6 and Bin7, but the number is dominated by Bin1.

### 3.2 Aerosol fluxes across the North Pacific

To better understand the source-receptor relationships, aerosol compositional mass flux exported from East Asia and imported to North America is estimated. The mass flux is estimated using zonal wind speed with a width of 10° in longitude, as illustrated in Figure 8. We calculate the aerosol mass flux from 20° N to 50° N centered at 130° E to represent the East Asia outflow, at 180° E to represent the North Pacific and at 130° W to represent North America inflow, respectively. On an annual basis, 48.7 Tg year$^{-1}$ of dust and 32.1 Tg year$^{-1}$ of pollution aerosols are exported from East Asia (20° N to 50° N), and 13.4 Tg year$^{-1}$ of dust and 10.3 Tg year$^{-1}$ of pollutions are imported into the west coast of North America. The model-estimated pollution aerosol mass flux is greater than the MODIS-estimated (about 14.0 Tg year$^{-1}$ exported from East Asia; 4.4 Tg year$^{-1}$ imported into North America) from Yu et al. (2008). Because this comparison is complicated by differences in the time period, the discrepancies may be due to the increasing pollution over East Asia under fast economic development. However, the dust mass flux is 60% smaller than the MODIS-CALIOP-estimated in 2008 (Yu et al., 2012). The discrepancy may be due to the different years and vertical distributions of dust and wind vector. Also, Yu et al., (2015) noted that there is a ±45%−70% dust mass flux uncertainty from the satellite-estimate.

For a more detailed dust comparison, the contributions of three major dust sources to the total aerosol mass are also analyzed (Fig. 8). The exported dust mass flux from EAS, NAF, and EIW are about 21.5, 15.2 and 12.1 Tg year$^{-1}$, which are about 26.5%, 18.8% and 14.9% to the total aerosol mass flux. After trans-Pacific transport, about 21.0% of EAS dust, 29.6% of NAF dust and 32.5% EIW dust arrive at the West Coast of the US where the NAF dust is the biggest contributor. Following dust, sulfate is another major composition in the exported and imported regions, and it contribute about 9.5 Tg year$^{-1}$ (11.7%) and 3.8Tg year$^{-1}$ (14.2%), respectively. Other polluting aerosols (e.g., nitrate, ammonium, organic matter, black carbon, other inorganic matter) are also exported from East Asia (about 16.9 Tg year$^{-1}$), transported across the North



Pacific (about 11.0 Tg year$^{-1}$), and imported into North America (about 6.5 Tg year$^{-1}$). Here, sea-salt is not included in the polluting aerosols as it is produced from North Pacific Ocean.

Previous studies showed that the seasonal variations of trans-Pacific transport aerosols are determined by the meteorological conditions, emissions, chemistry and wet/dry deposition processes (Yu et al., 2008; Hu et al., 2016).
However, few studies focused on the seasonal variations of aerosol compositions along the trans-Pacific pathway. Figure 9 shows the seasonal variations of meridionally integrated aerosol component mass flux for 2010–2014 across East Asia (130° E), North Pacific (180°) and North America (130° W), contributed by dust, sulfate, nitrate, ammonium, organic matter, black carbon, other inorganic matter and sea-salt. For the East Asia outflow, the highest aerosol mass flux of 39.8 Tg year$^{-1}$ is within the 30° N – 40° N segment, followed by 32.0 Tg year$^{-1}$ in the 40° N – 50° N segment and 9.3 Tg year$^{-1}$ in the 20° N –
30° N segment. The NAF dust (1.8 Tg year$^{-1}$) dominates the total aerosol mass in the 20° N – 30° N sub-tropical segment, and the EAS dust (12.2 Tg year$^{-1}$) dominates in the 40° N – 50° N segment. In the 30° N – 40° N segment, the EAS dust, NAF dust and EIW dust contribute about 8.7, 8.3 and 6.8 Tg year$^{-1}$, respectively. The total aerosol mass flux peaks in MAM over the three meridional segments followed by DJF, because of stronger springtime dry convection over East Asia (Dickerson et al., 2007), and stronger and more frequent warm conveyor belts (WCB) in spring and winter (Eckhardt et al.,
2004). The dust (about 25.6 Tg) exported from East Asia can contribute up to 63.1% to the total aerosol mass concentration in MAM, in which the EAS dust (about 11.4 Tg) is the greatest contributor, about 28.1%. In addition to dust, the aerosols are mainly composed of anthropogenic pollution and biomass burning particles, about 36.3% (14.7 Tg), in which the OIN is the greatest contributors, about 10.0% (4.1 Tg). For example, the maximum BC and OC in MAM indicates that the strongest biomass burning occurs in MAM (Giglio et al., 2006; Bian et al., 2007).
Over the North Pacific region, the aerosol mass flux reduces to about 38.5% (DJF), 42.3% (MAM), 47.6% (JJA) and 43.3% (SON), compared with aerosols from East Asia. In DJF, the NAF dust (about 2.2 Tg) and EIW dust (about 1.5 Tg) are significantly greater than the EAS dust (about 0.8 Tg). The decreasing EAS dust could be attributed to its low altitude, so it can be removed much easier by rain. Sea-salt is in the North Pacific is more abundant than in other sub-regions, with a maximum value in DJF.
For the west coast of North America inflow, the aerosol mass flux accounts for about 8.8% (0.8 Tg year$^{-1}$), 26.3% (10.4 Tg year$^{-1}$) and 48.5% (15.5 Tg year$^{-1}$) in the 20° N – 30° N, 30° N – 40° N and 40° N – 50° N segments compared with the outflow. Aerosol mass flux in the 40° N – 50° N segment is greater than that in the 30° N – 40° N segment because of the poleward shift of aerosols during the trans-Pacific transport (Yu et al., 2008). As the dominating aerosol composition, trans-Pacific dust is about 7.6 Tg year$^{-1}$ in the 40° N – 50° N segment, in which the EAS dust, NAF dust and EIW dust is about
2.8, 2.8 and 2.1 Tg year$^{-1}$, respectively. Also, we can see that larger change of aerosol mass flux occurs in the 30° N – 40° N segment, and smaller change occurs in the 40° N – 50° N segment. In the 20° N – 30° N segment, aerosols from North America have a westward transport (negative flux) in JJA and SON, because of the westward transport of sea-salt aerosol. This phenomenon is also shown by Yu et al. (2008). Overall, 36.1% (7.6 Tg year$^{-1}$) of the trans-Pacific transport dust and 51.3% (5.4 Tg year$^{-1}$) of pollution aerosols arrive in North America in the 40° N – 50° N segment. For the 30° N – 40° N



segment, the contributions are 21.1% (dust, 5.0 Tg year$^{-1}$) and 26.3% (pollutions, 4.0 Tg year$^{-1}$), respectively. In MAM and JJA, dust is the major contributor followed by sulfate. However, in DJF and SON, sea-salt is the major contributor followed by dust. For the pollution aerosols, sulfate is the major contributor, which is about 3.8 Tg year$^{-1}$ (14.2% to total aerosol mass).

In general, aerosols between the 30° N – 50° N segment contribute about 48.8% of dust to the total aerosol mass. The pollution aerosol contribution is smaller in the 30° N – 50° N segment compared with dust, but it is opposite in the 20° N – 30° N segment. This is attributed to outflow aerosols occurring mainly in the 30° N – 50° N segment, especially associated with dust (Fig. 1). Also, the smallest mass concentration in the 20° N – 30° N segment is likely to result from the shift of westerlies to easterlies in the free troposphere (Yu et al., 2008). Significantly, the dust in the 20° N – 30° N segment from

Middle East and Africa is about two times of the dust from East Asia. During the transport over Pacific Ocean, the aerosols rapidly decrease in the 20° N – 40° N segment because of wet deposition induced by precipitation (Hu et al., 2016). The annual mean of dust flux inflow to North America is about 13.4 Tg year$^{-1}$, and the proportion is about 50.1%. This is followed by sulfate with a flux of about 3.8 Tg year$^{-1}$ (14.2%). It is obvious that dust is the major natural aerosols and sulfate is the major anthropogenic aerosols in the trans-Pacific transport.

**3.3 Aerosol direct radiative forcing**

The spatial distribution of aerosol compositional direct radiative forcing averaged for 2010–2014 at TOA, in the atmosphere (ATM), and at the SFC under all-sky conditions is shown in Figure 10. The spatial distribution of aerosol compositional direct radiative forcing closely follows the corresponding aerosol compositional mass. At the SFC, all aerosol compositions reduce direct radiative fluxes and result in cooling effect from the trans-Pacific transport. It is interesting to note that the

black carbon mass is relatively small (about 0.83% contribution to total aerosol mass), but it causes larger cooling effect at the SFC, especially over India and Southeast China with the largest forcing value of –8 W m$^{-2}$. The maximum dust direct radiative forcing at the SFC is dominated over Taklimakan, Gobi Desert, and Arabian Sea with negative value of about –6 W m$^{-2}$. This pattern is consistent with the result from Zhao et al. (2013b) and is inside the range of –5.2 ~ –15.6 W m$^{-2}$ from Bi et al. (2013) and Huang et al. (2014). In the ATM, aerosol compositions lead to a warming effect, with black carbon

inducing the largest warming of about 8 W m$^{-2}$, and the dust and sulfate direct radiative forcing is surprisingly much small even though they have larger mass. This can be attributed to the strong absorbing property of black carbon. Dust produces a warming effect with a maximum value of about +2.0 W m$^{-2}$ and a domain average of +0.13 W m$^{-2}$ in the ATM. At the TOA, dust, sulfate, organic matter and other aerosol result in cooling, but black carbon results in warming with the highest value of +8.0 W m$^{-2}$ over India and Southeast China. Overall, black carbon results in a significant cooling effect at the SFC and

warming effect in the ATM and at the TOA. Dust also causes a significant cooling effect at the SFC and TOA.

Figure 11 shows the seasonal variation of aerosol direct radiative forcing over the three sub-regions (trans-Pacific transport path) under all-sky conditions. The direct radiative forcing of the five aerosol compositions is large in MAM due to high concentrations of pollution aerosols and dust along the trans-Pacific transport pathway. The forcing maximum is



produced by black carbon especially in the ATM, even though black carbon mass is lowest in the total aerosol mass. The seasonal variations of direct radiative forcing are consistent with that of the aerosol column mass (Fig. 9). Over the western Pacific, black carbon results in a surface cooling of $-4.43 \sim -1.84$ W m$^{-2}$, atmosphere warming of $+2.61 \sim +6.80$ W m$^{-2}$, and TOA cooling of $+0.76 \sim +2.38$ W m$^{-2}$. Dust results in a surface cooling of $-3.10 \sim -0.92$ W m$^{-2}$, atmosphere warming of

$+0.02 \sim +0.61$ W m$^{-2}$, and TOA cooling of $-2.48 \sim -0.91$ W m$^{-2}$. During the trans-Pacific transport, the direct radiative forcing rapidly decreases as aerosols are deposited. Over the eastern Pacific, BC results in a surface cooling of $-1.79 \sim -0.42$ W m$^{-2}$, atmosphere warming of $+0.61 \sim +2.75$ W m$^{-2}$, and TOA cooling of $+0.19 \sim +0.96$ W m$^{-2}$. Dust results in a surface cooling of $-1.72 \sim -0.34$ Wm$^{-2}$, atmosphere warming of $+0.005 \sim +0.13$ Wm$^{-2}$, and TOA cooling of $-1.59 \sim -0.32$ W m$^{-2}$. The annual mean direct radiative forcing of aerosols over the three sub-regions is also shown in table 1. The trans-Pacific

transport aerosols result in a surface cooling of $-3.68$ W m$^{-2}$, atmosphere warming of $+1.40$ W m$^{-2}$, and TOA cooling of $-2.28$ W m$^{-2}$.

### 3.4 Impact of trans-pacific transport over the US

### 3.4.1 Mass versus number concentrations

Aerosols can reach and influence the North American environment and regional climate through trans-Pacific transport

(Chin et al., 2007; Yu et al., 2012; Huang et al., 2014; Hu et al., 2016). Figure 12 shows the spatial distribution of source contribution of trans-Pacific transport and North American local aerosol mass integrated in the atmospheric column and at the surface. To better describe the contribution from the two sources, North America is divided into three sub-regions: West (130° W – 110° W), Central (110° W – 90° W), and East (90° W – 70° W). Clearly, the aerosol mass over the ocean near North America is dominated by the trans-Pacific transport (more than 90% contribution). However, over the continent,

aerosols from the North American have significant contributions. In the atmospheric column, aerosols from trans-Pacific transport have a larger contribution than those from the US, except for the Nevada desert region, where a large amount of dust can be emitted. Over this desert region, the local source contribution is about 60% compared to about 40% from trans-Pacific transport. Other than this region, the continental North American is mainly influence by transported aerosols (about 50–60%). Unlike the source contribution of column aerosol mass, the surface aerosol mass contribution mainly comes from

local emission, which can reach more than 60%. Hence surface air quality over North America is mainly influenced by local sources including pollution aerosols and dust. At the higher altitude, the transported aerosols are the major particles, which can influence clouds (Zhao et al., 2012) and radiative forcing (Yu et al., 2012).

    Figure 13 shows the seasonal and annual variation of trans-Pacific transport and North American total aerosol mass fractional in the column and at the surface over the three sub-regions of North America. For the West, pollution aerosols and

dust from transport have important contributions in the column (more than 60%). At the surface, sea-salt is dominated aerosols from transport, followed by local dust, with similar fraction seasonally. The annual mean of dust in the column is about 34.7% and sea-salt at the surface is about 44.6%. Further away from the ocean, the influence of sea-salt is reduced in



the Central region. The transported aerosols are about 65.4% (in column) and 36.8% (at surface) of the total, with a peak in MAM of 73.3% (in column) and 41.7% (at surface). At the surface, we also can see that more than 35% of OIN and nitrate are contributed by the local source with the seasonality of nitrate opposite to that of sulfate. The maximum nitrate mass occurs in the cold season (DJF), because of the combined effects of vertical turbulent mixing and temperature (Zhao et al.,

2013a; Hu et al., 2016). For the East, the aerosol contribution is similar with the West in the column. However, at the surface, there is little dust contribution from local source year round, and the NAF dust dominates the transported aerosols through trans-Atlantic transport, especially in JJA (Yu et al., 2015). The NAF dust contributes about 27% (except sea-salt) to the column and surface. The trans-Atlantic transport of dust is not discussed in this study, because it has been studied in most previous works (e.g., Dunion and Velden, 2004; Mahowald et al., 2008; Wilcox et al, 2010; Yu et al., 2015)

Figure 14 shows the seasonal and annual variation of trans-Pacific transport and North American total fractional aerosol number concentration in the column and at the surface. In the column, long-range transport aerosols have significant influence on the aerosol number concentration over the US. The maximum contribution is 80% over the West, followed by 52.7% over the Central and 51.4% over the East. However,the small particles are above the planetary boundary layer, so the maximum aerosol number concentration occurs in DJF at the surface, which is consistent with the aerosol mass. At the

surface, the trans-Pacific aerosols contribute about 44.3% of number concentration over the West, followed by 17.2% over the Central and 16.3% over the East. Also, we can see that nitrate from local sources has a maximum contribution in DJF and minimum contribution in JJA. In general, surface aerosol particles are mainly from local emissions over the Central and East, especially for pollution aerosols. This is consistent with a previous study claiming that local emissions of North America dominated the eastern US surface pollution particles (Chin et al., 2007).

**3.4.2 Aerosol direct radiative forcing**

Figure 15 shows the seasonal and annual variation of fractional contribution from trans-Pacific (red) and North American local emitted (blue) aerosols to the total direct radiative forcing at the TOA, in the ATM, and at the SFC. The contribution from trans-Pacific aerosols has seasonal variation with a peak in MAM due to the strongest transport of aerosols. Also, the trans-Pacific aerosol produces the largest forcing effect over the western North American (dominated by trans-Pacific

transport), followed by the eastern North American (dominated by trans-Atlantic transport). Overall, aerosols result in TOA cooling, atmosphere warming and surface cooling. Over the West region, because of the local dust aerosol with little forcing, the transported aerosol can produce annual mean direct radiative forcing of –2.91 (Surface, 86.4%), +1.36 (atmosphere, 85.5%) and –1.55 (TOA, 87.1%) W m$^{-2}$. The minimum total direct radiative forcing occurs in JJA at the TOA, when strong wet deposition results in aerosol mass reduction. The direct radiative forcing ranges from –2.99 to –0.87 Wm$^{-2}$ from

transported aerosols and –0.43 to –0.10 Wm$^{-2}$ from North American aerosols. In the ATM, the forcing ranges from +0.57 to +2.69 W m$^{-2}$ and +0.10 to +0.29 W m$^{-2}$ for the transported and North American aerosols, respectively. Generally, the forcing at the surface is larger, which ranges from –5.68 to –1.48 Wm$^{-2}$ from the transported aerosols and –0.72 to –0.20 W m$^{-2}$ from the North American aerosols. Compared with the western North American, the eastern US has more pollution aerosols



emitted locally and less aerosols transported remotely (Chin et al., 2007). Therefore, Figure 15 shows a significant increase of the direct radiative forcing from North American aerosol over the Central and East regions. The pattern is similar with a peak in MAM at TOA and Surface, but for the atmosphere a peak occurs in JJA. The direct radiative forcing induced by the transported aerosols ranges from –3.98 to –1.11 W m$^{-2}$ (–4.30 to –1.02 W m$^{-2}$) at the SFC, +0.37 to +2.48 W m$^{-2}$ (+0.54 to +2.25 W m$^{-2}$) in the ATM, and –2.04 to –0.74 W m$^{-2}$ (–2.04 to –0.48 W m$^{-2}$) at the TOA over the East (Central) region.

## 4 Summary and conclusion

In this study, the characteristics of trans-Pacific transport of aerosols and their source contributions are investigated using an updated version of WRF-Chem. To better describe the aerosols inter-continental transport, quasi-global simulations (180$^{\circ}$ W – 180$^{\circ}$ E and 70$^{\circ}$ S – 75$^{\circ}$ N) are conducted for the period of 2010–2014 to capture the spatial distribution and seasonal variation of both mass and number for various aerosol species. The fluxes of mass composition and direct radiative forcing of aerosols during the trans-Pacific transport are further discussed. Finally, the source contribution of aerosol composition mass, number concentration, and direct radiative forcing are quantified over the US. Generally, the updated model can capture the spatiotemporal characteristics of precipitation, wind, aerosol extinction profiles, AOD, EAE, AAOD, and surface aerosol mass concentrations compared with various reanalysis and observation data. Our results further show that the model can reproduce the spatiotemporal characteristics of trans-Pacific aerosols and the associated impacts on the regional surface air quality over the West Coast of the US (Hu et al., 2016). Moreover, the WRF-Chem simulation using 8-BIN can better reproduce the distribution of aerosol size and number (Zhao et al., 2013b), giving more detailed information about the spatial distribution (horizontal and vertical) of aerosols and the aerosol trans-Pacific transport process. Quantitative analysis of aerosol source contribution and direct radiative forcing provides more evidence for the impacts of aerosols transported from eastern Asia on the regional air quality and climate of the western US. Finally, the tracer-tagging technique (Wang et al., 2014) is shown to be a useful tool to study the contributions of dust particles emitted from four main desert sources (North America: 140$^{\circ}$ W – 80$^{\circ}$ W, 15$^{\circ}$ N – 50$^{\circ}$ N; East Asia: 75$^{\circ}$ E – 150$^{\circ}$ E, 25$^{\circ}$ N – 50$^{\circ}$ N; North Africa: 20$^{\circ}$ W – 35$^{\circ}$ E, 0 – 40$^{\circ}$ W; and Middle East and Central Asia: other regions) and their contribution during transport of individual dust emissions over Pacific Ocean.

Aerosol mass exported from East Asia are dominated by dust particles, in which dust aerosols from the North African and Middle East and Central Aisa are carried to East Asia at high altitude (above 4 km) and mixed with dust from the Taklimakan and Gobi deserts, then transport across the Pacific Ocean. The East Asian dust contributes about 28.3% (22.3 mg m$^{-2}$) to the total aerosol mass concentration, followed by North Africa dust (11.0%, 8.7 mg m$^{-2}$) and East Asia dust (9.0%, 7.1 mg m$^{-2}$). During the trans-Pacific transport, the aerosol mass concentration gradually decreases with higher concentration over East Asia (about 350 mg m$^{-2}$) and dominates the total aerosol mass north of 30$^{\circ}$ N over the North Pacific due to strong westerly winds. Generally, the maximum aerosol mass concentration occurs below 4 km with about 15 $\mu$g m$^{-3}$ in the 27$^{\circ}$ N – 44$^{\circ}$ N segment in spring. Sea-salt dominates the total aerosol mass along the Pacific Ocean below 1 km and





dust dominates the total aerosols above 4 km. Also, the aerosol plumes have an obvious poleward shift by the easterlies wind during the trans-Pacific transport. When imported into the western North America, the transported dust contributes 41.1% to the total aerosol loading, followed by sea-salt of about 21.7%. The dust mass concentration from North America is 0.48 mg m$^{-2}$, which is far lower than the East Asian dust (EAS, 4.8 mg m$^{-2}$), North African dust (NAF, 4.6 mg m$^{-2}$) and Middle East

and Central Asia dust (EIW, 3.6 mg m$^{-2}$). Our results suggest that trans-Pacific transport dust as the major aerosol component from North Africa desert, Middle East and Central Asia desert, and East Asia desert has similar contribution.

Similar to aerosol mass, aerosol number also decreases along the Pacific Ocean, with a peak in spring. Different from the mass composition of dust, the number concentration is dominated by sulfate. The aerosols with size 0.039~0.156 $\mu$m (Bin1-Bin3) are the main number contributors during transport, but aerosol mass is dominated by the size range of 2.5~10.0 $\mu$m

below 2 km, with a maximum at 5.0 $\mu$m (about 40%). Above 2 km, the aerosol size of 1.25~5.0 $\mu$m becomes the main mass distribution range with more than 40% contribution. These aerosol particles can reach mid-to-upper troposphere during the inter-continent transport (Fig. 6 and Fig. 7), and could provide an important source of ice nuclei and influence the cloud radiative forcing.

While this study shows that about 80.8 Tg year$^{-1}$ aerosols are exported from East Asia (20° N to 50° N) with a peak (40.6

Tg) in spring, only about 26.7 Tg year$^{-1}$ of aerosols arrive in the west coast of North America with a peak (15.0 Tg year$^{-1}$) in spring. Over West of North America, about 50–60% of column aerosols is from trans-Pacific transport, except for the Nevada desert region, where dust aerosol from local deserts can contribute more than 60%. Different from the transported aerosol contribution to the column and high-altitude atmosphere, surface aerosol is mainly from local areas with more than 60% contribution in spring. Over all, dust contributes about 34.7% to the annual mean aerosol in the column, and sea-salt is

about 44.6% at the surface. Over the Central and East regions, the influence of local emission increases significantly, with OIN and nitrate aerosols contributing more than 30%. Furthermore, dust from Africa is dominated by the transported aerosol through the Atlantic Ocean, especially in JJA and consist with Yu et al. (2015).

The impact of aerosols on regional climate can be characterized by the magnitude of radiative forcing. Our calculations show that aerosols can induce a warming and cooling effect respectively in the atmosphere and at the surface. At the TOA,

aerosols produce a cooling effect, except for BC aerosols with a warming effect. It is interesting to note that BC, with smallest mass, can produce the largest direct radiative forcing, especially over polluted aerosol source regions (e.g., India, Southeast of China). Because of the strongest pollution aerosols and dust transport, the largest direct radiative forcing occurs in spring. For the North America region, the transported aerosols have an annual mean direct radiative forcing of –2.65 (at the SFC, 87.1%), +1.41 (in the atmosphere, 87%) and –1.52 (at TOA, 87.3%) W m$^{-2}$ over the western North American.

Based on the WRF-Chem simulations of aerosol trans-Pacific transport, the aerosol characteristics and source contributions are analyzed. Modeling studies with reliable representations of aerosol transport mechanisms show that dust and pollution have significant impact on the surface air quality and radiative forcing in North America. However, the feedback on the trans-Pacific transport of dust and pollution from the changes in atmospheric circulations is not clearly (Yu et al., 2008). Long-term observations are needed to further evaluate the modeling results due to year-to-year variations of





dust and increase of biomass burning emissions as the economic growth. Long-term modeling results may help study the interaction between transported aerosols and climate and better understand the impact of trans-Pacific aerosols on regional climate.

**Code availability**

The release version 3.5.1 of WRF-Chem can be download from http://www2.mmm.ucar.edu/wrf/users/download/get_source.html. The updated model is available to contact the first author (huzy@lzu.edu.cn). Also, the code modifications will be incorporated the release version of WRF-Chem.

**Author contributions**

Zhiyuan Hu and Chun Zhao conducted the quasi-global simulations. Zhiyuan Hu performed the analyses, wrote the paper and coordinated the paper. All authors contributed to the final version of the paper.

**Acknowledgements**

This research was supported by the Foundation for National Natural Science Foundation of China (No. 41805116) and the Fundamental Research Funds for the Central Universities lzujbky-2018-49, Innovative Research Groups of the National

20 Science Foundation of China (grant no. 41521004) and Strategic Priority Research Program of Chinese Academy of Sciences, (Grant No. XDA20060103). Chun Zhao was supported by the "Thousand Talents Plan for Young Professionals" program and the National Natural Science Foundation of China NSFC (Grant No. 41775146) of China. Yun Qian and L. Ruby Leung were supported by Office of Science, US Department of Energy Biological and Environmental Research, through the Regional and Global Modeling and Analysis program. PNNL is operated by Battelle Memorial Institute for the

25 US Department of Energy under contract DE-AC06-76RLO–1830.



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



**Table 1.** Annual mean radiative forcing of aerosols simulated by WRF-Chem for 2010-2014 over three regions shown in Fig.

2. Negative values represent downward radiation. Units: Wm$^{-2}$.

|  | TOA | ATM | SFC |
| --- | --- | --- | --- |
| the western Pacific | -4.08 | 5.36 | -9.44 |
| the central Pacific | -2.82 | 2.07 | -4.89 |
| the eastern Pacific | -2.28 | 1.40 | -3.68 |





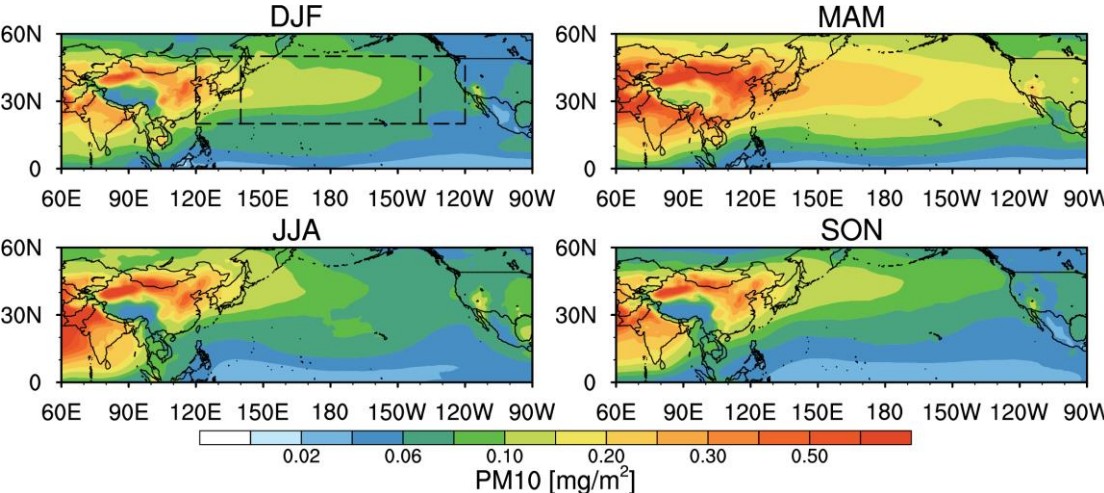

**Figure 1.** Spatial distribution of seasonal mean aerosol column mass loading from WRF-Chem averaged for 2010-2014.

Three regions are denoted by the black boxes: the western Pacific (20° N – 50° N and 120° E – 140° E), the central Pacific

(20° N – 50ºN and 140° E – 140° W), and the eastern Pacific (20° N – 50° N and 140° W –120° W) for analysis.



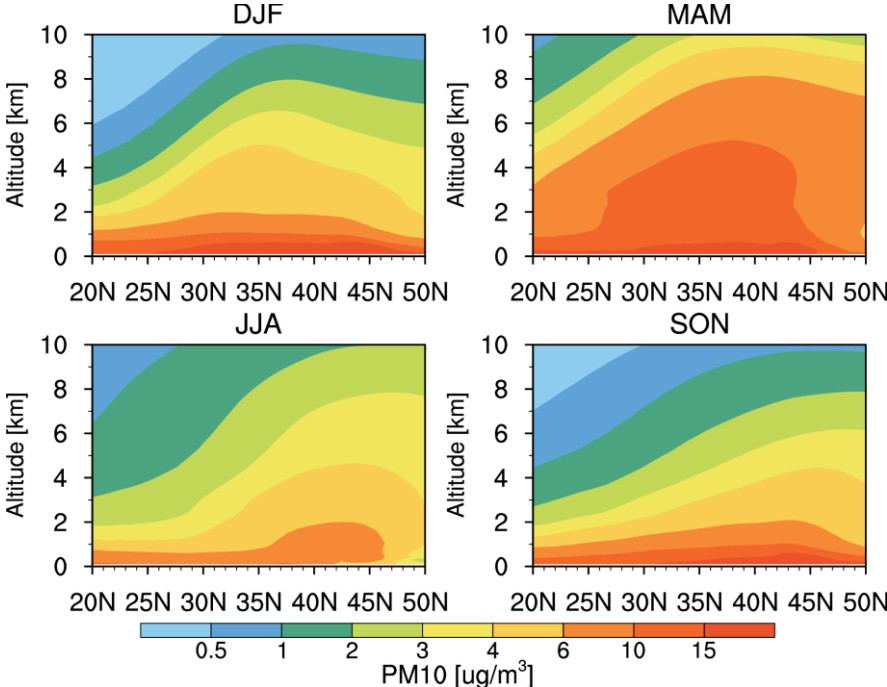

**Figure 2** Vertical cross-section of zonal mean aerosol mass concentration averaged for each season from the WRF-Chem simulation averaged for 2010-2014.



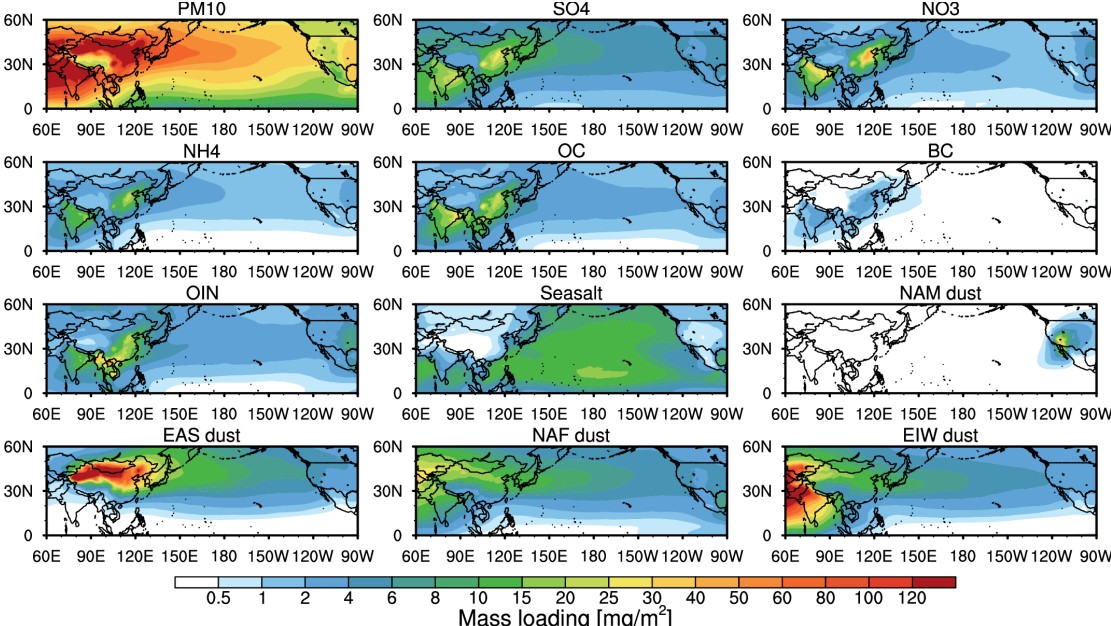

**Figure 3** Spatial distribution of aerosol composition column mass loading from WRF-Chem averaged for 2010-2014. The

5    trans-Pacific transport aerosol mass spatial distribution is denoted by PM10 and the dust from North America, East Asia,

North Africa, elsewhere in the world are denoted by NAM dust, EAS dust, NAF dust, EIW dust.





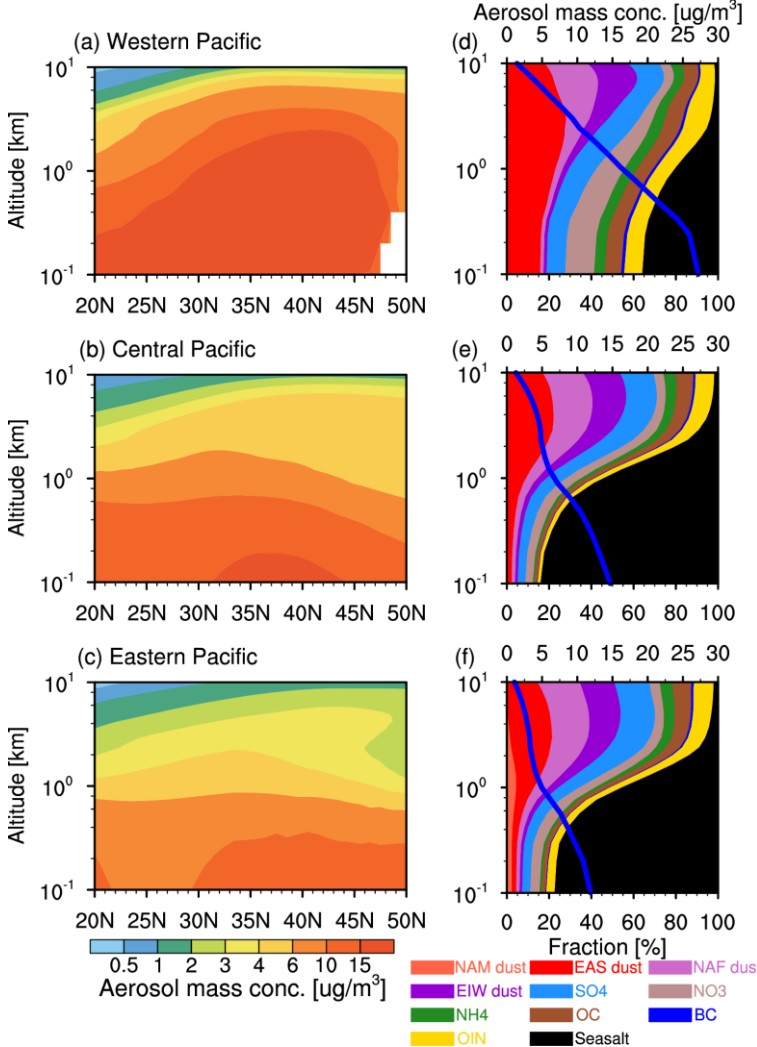

**Figure 4.** Vertical cross-section of zonal mean aerosol mass concentration and vertical distributions of mean aerosol mass

5   (blue solid line) and the composition fractions (colored shade-contour) from the WRF-Chem simulation averaged for 2010-

2014 over the three regions shown in Fig. 1.



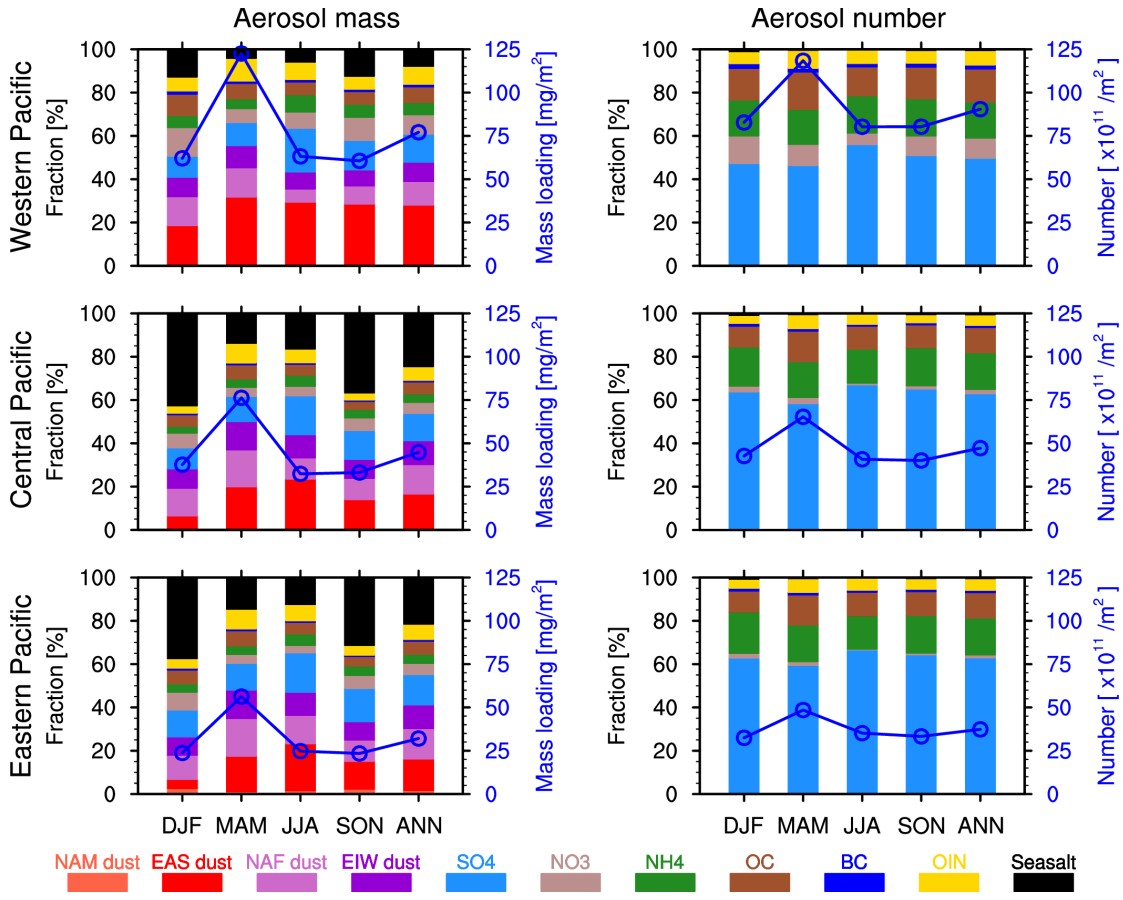

**Figure 5.** Fractional contributions for seasonal and annual mean aerosol mass and number over the western, central and

eastern Pacific from the WRF-Chem simulation averaged for 2010-2014.



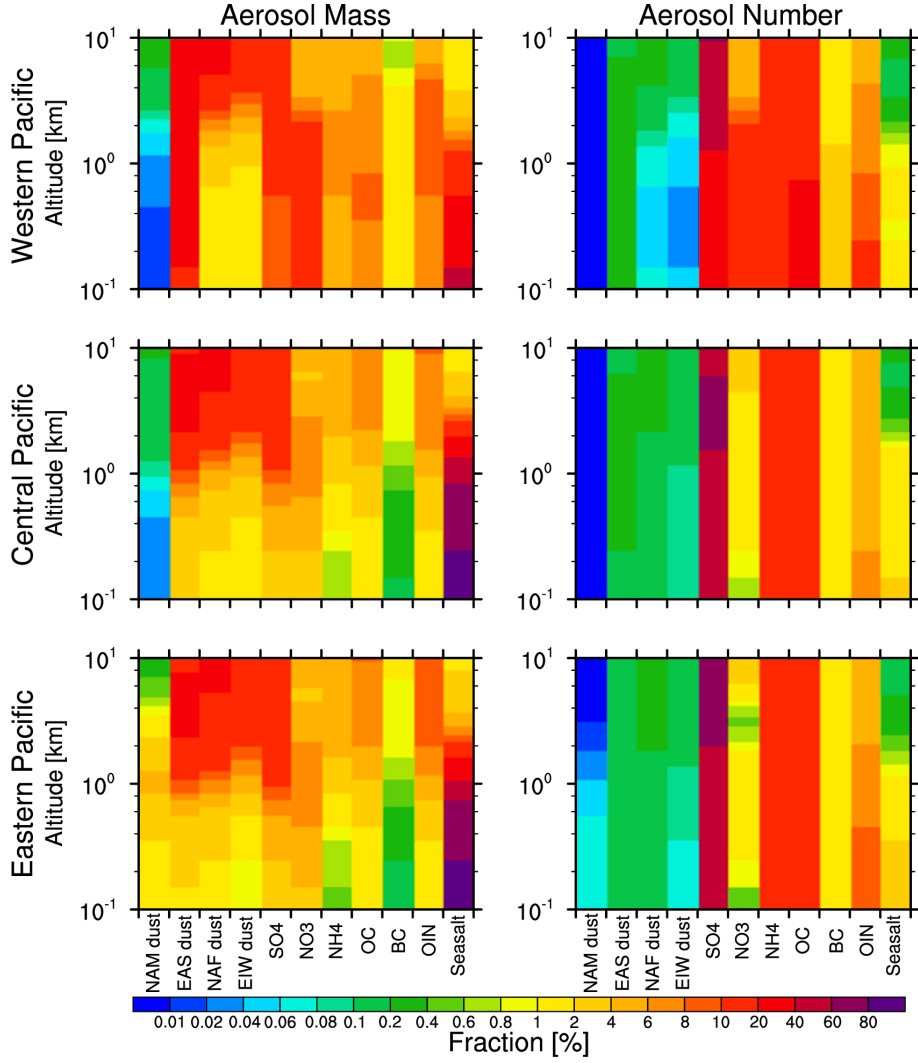

**Figure 6.** Vertical distribution of fractional contributions to aerosol composition mass and number over the western, central

5   and eastern Pacific from the WRF-Chem simulation averaged for 2010-2014.



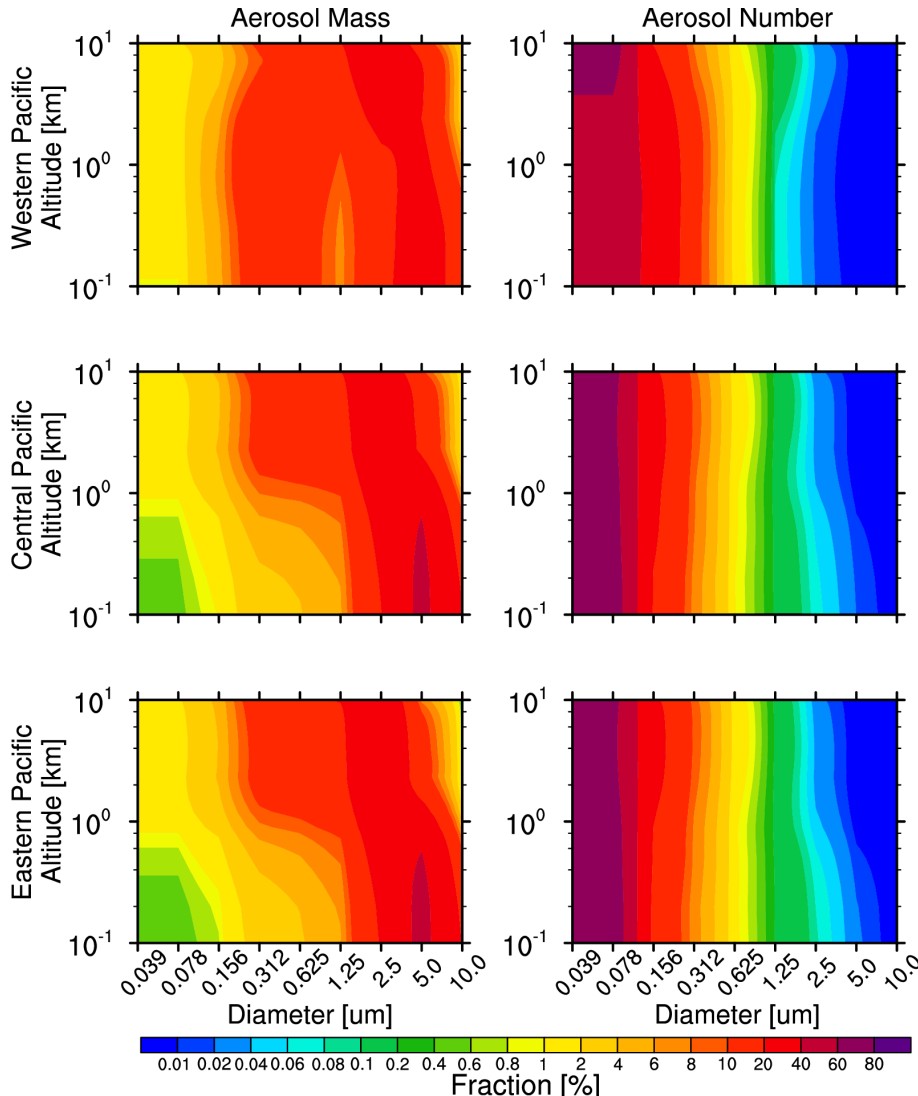

**Figure 7.** Vertical distribution of fractional contributions to the size of aerosol mass and number over the western, central

5    and eastern Pacific from the WRF-Chem simulation averaged for 2010-2014.



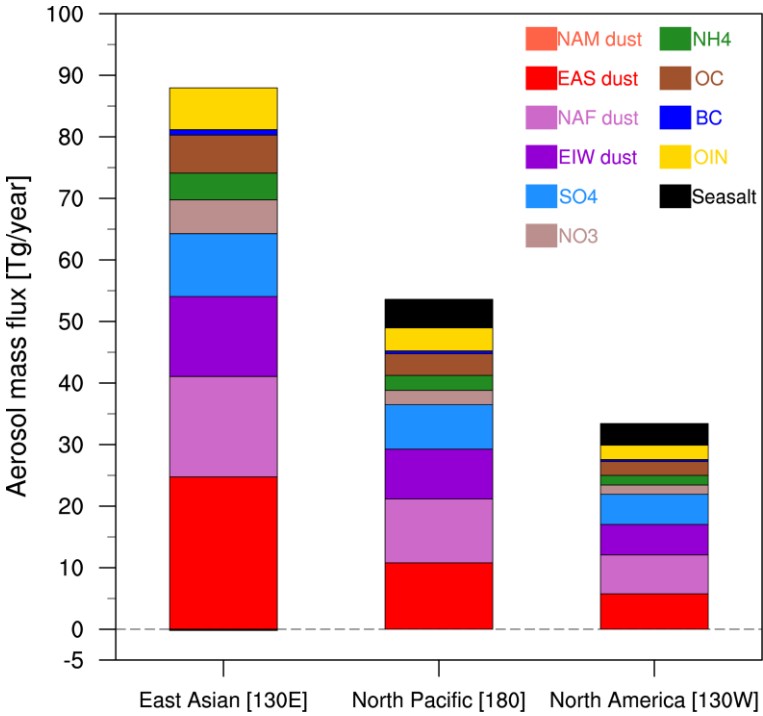

**Figure 8.** Model-based estimate of aerosol annual mass flux in East Asian outflow, across the North Pacific and North America inflow from the WRF-Chem simulation averaged for 2010-2014.





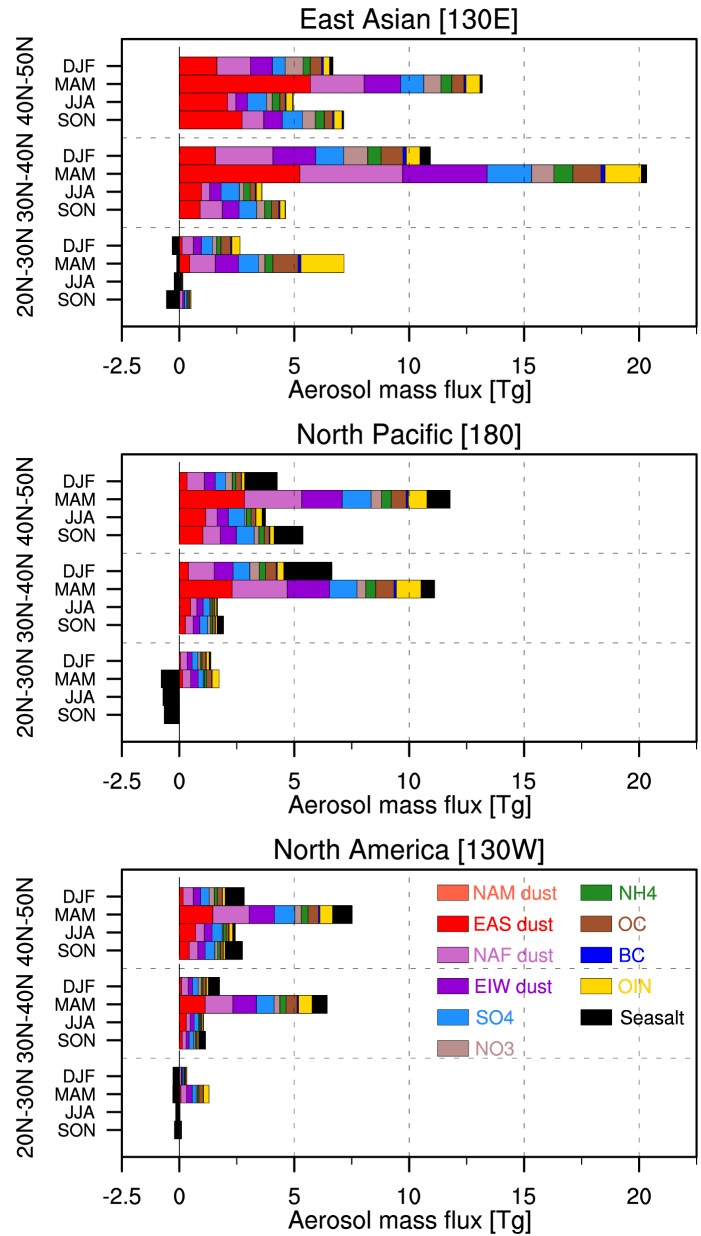

**Figure 9.** Meridional variations of East Asia outflow, across the North Pacific and North America inflow of aerosol seasonal mass flux for 2010-2014.





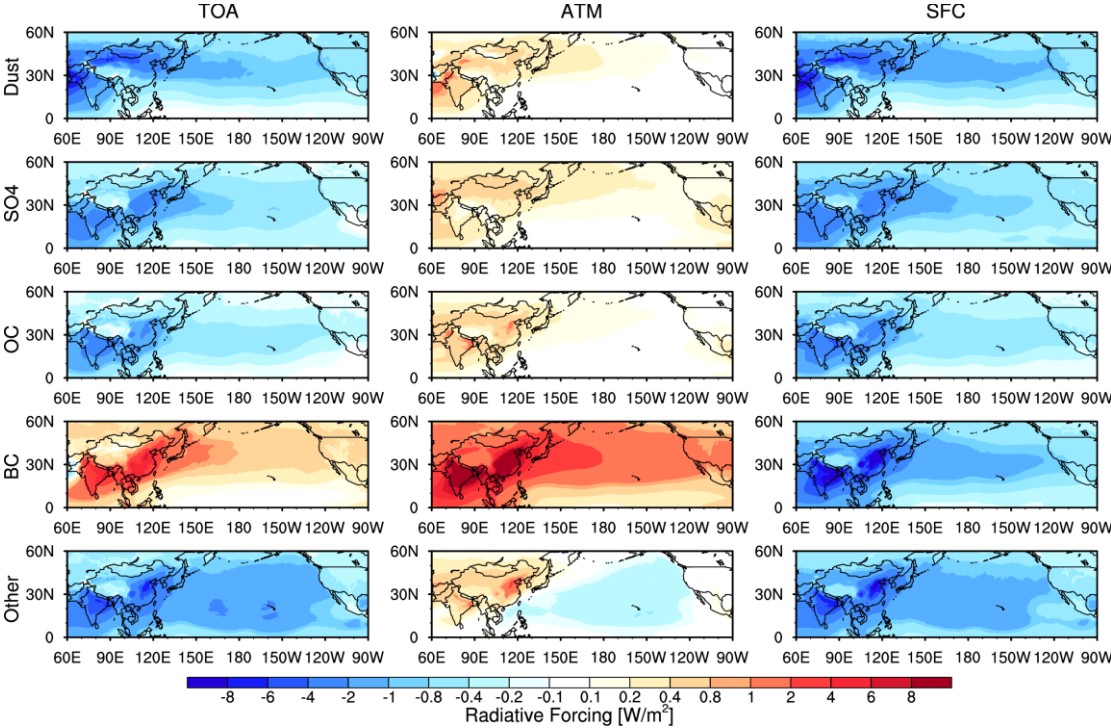

**Figure 10.** Spatial distribution of aerosol composition direct radiative forcing at the TOA (left), in the ATM (middle) and at
the SFC (right) under all-sky conditions from the WRF-Chem simulation averaged for 2010-2014. The Other compositions
induce ammonium, nitrate, sea-salt, and unspeciated $PM_{2.5}$.





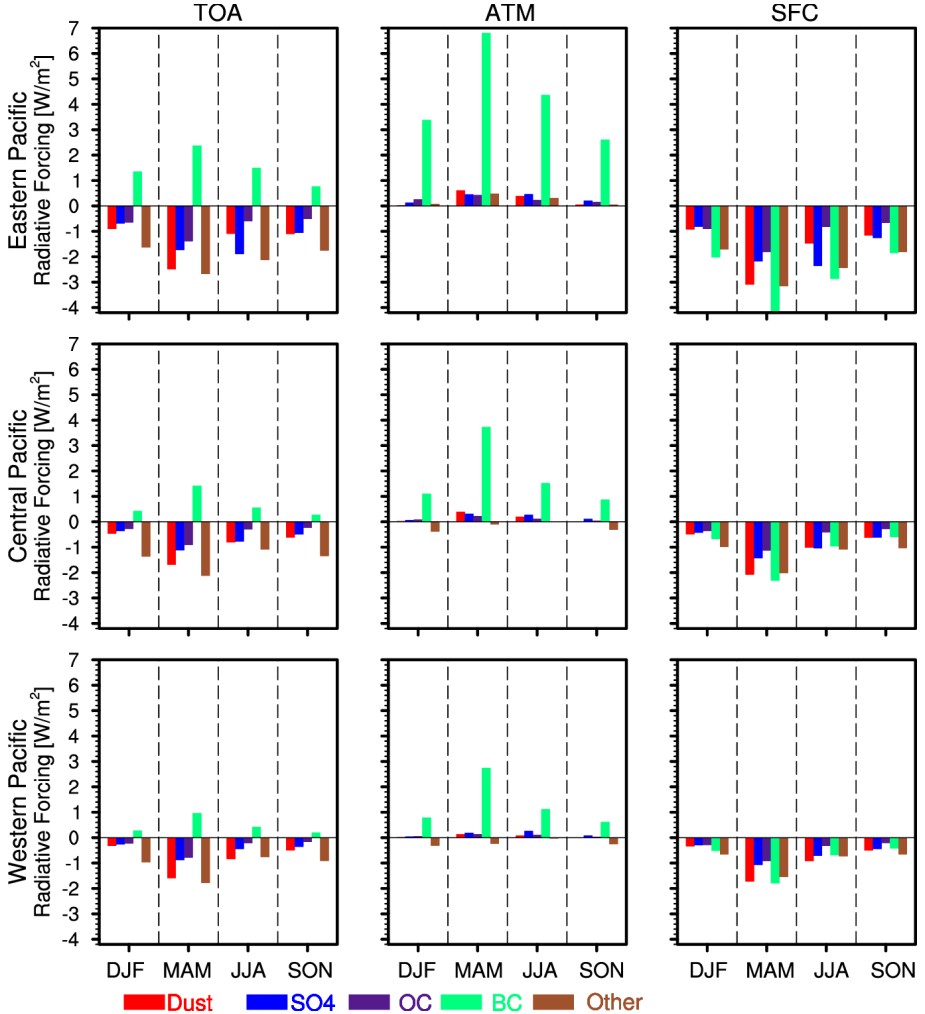

**Figure 11.** Seasonal variation of aerosol composition radiative forcing at the TOA (left), in the atmosphere (middle) and at

the surface (right) over the western, central and eastern Pacific under all-sky conditions from the WRF-Chem simulation

averaged for 2010-2014.





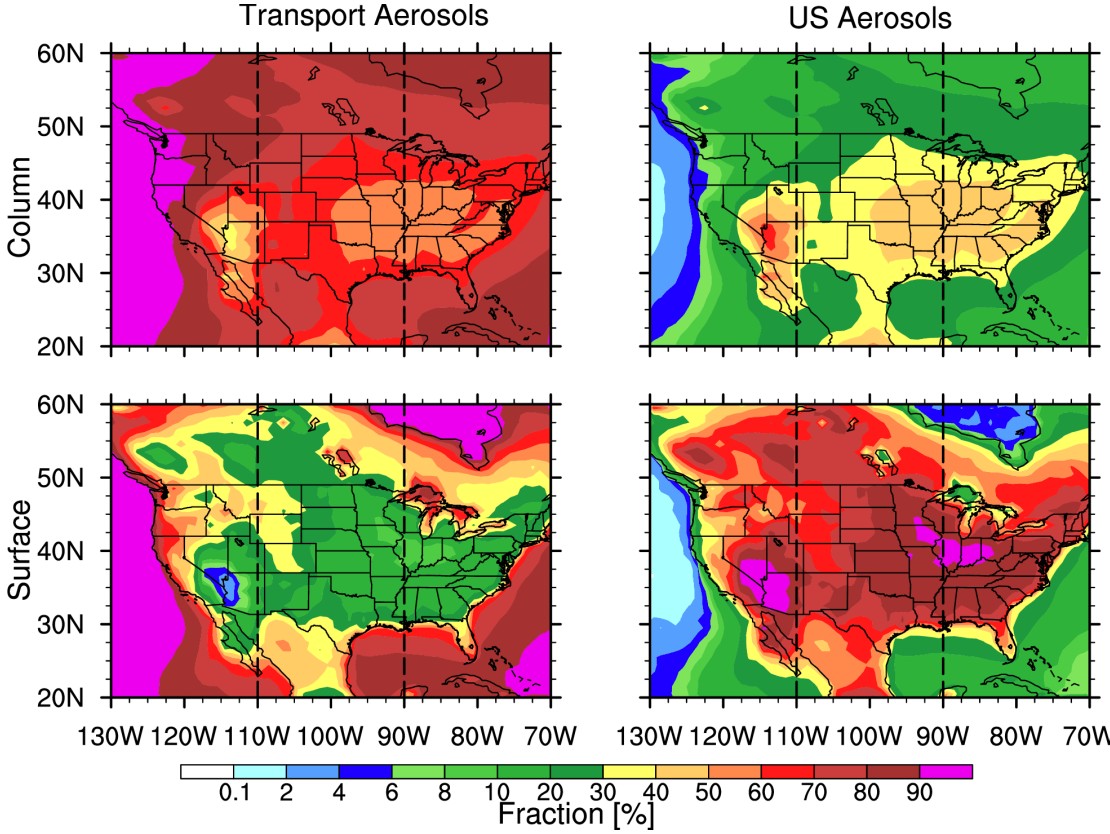

**Figure 12.** Spatial distribution of trans-Pacific transport and US total aerosol mass fraction for the column and at the surface from the WRF-Chem simulations averaged for 2010-2014. The North American region is divided into three sub-regions bounded by the dotted lines: West (20° N – 60° N and 130° W – 110° W), Central (20° W – 60° N and 110° W – 90° W), and East (20° N – 60° N and 90° W – 70° W).





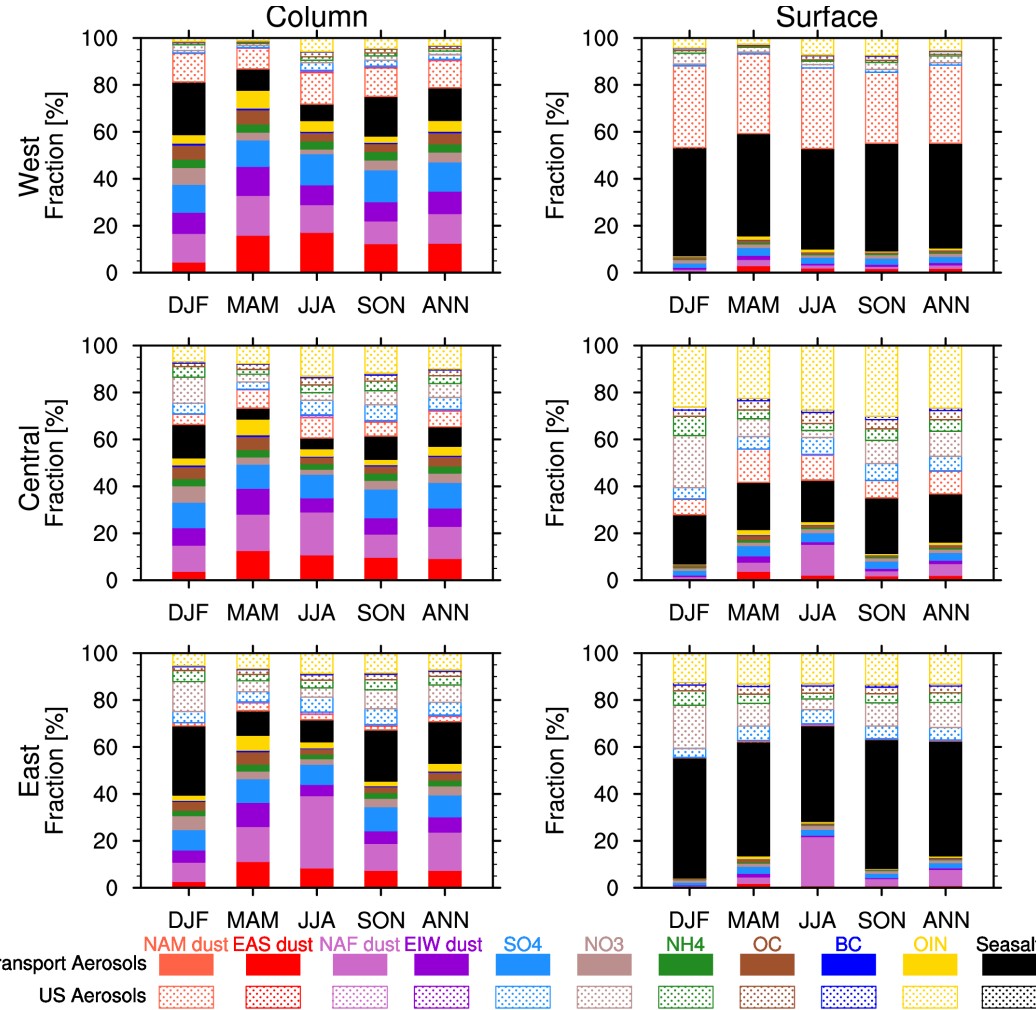

**Figure 13.** Seasonal and annual variation of trans-Pacific transport and US total aerosol mass fractional contribution for the

5   column and at the surface from the WRF-Chem simulations averaged for 2010-2014 in the western, central and eastern

North America shown in Fig.12.





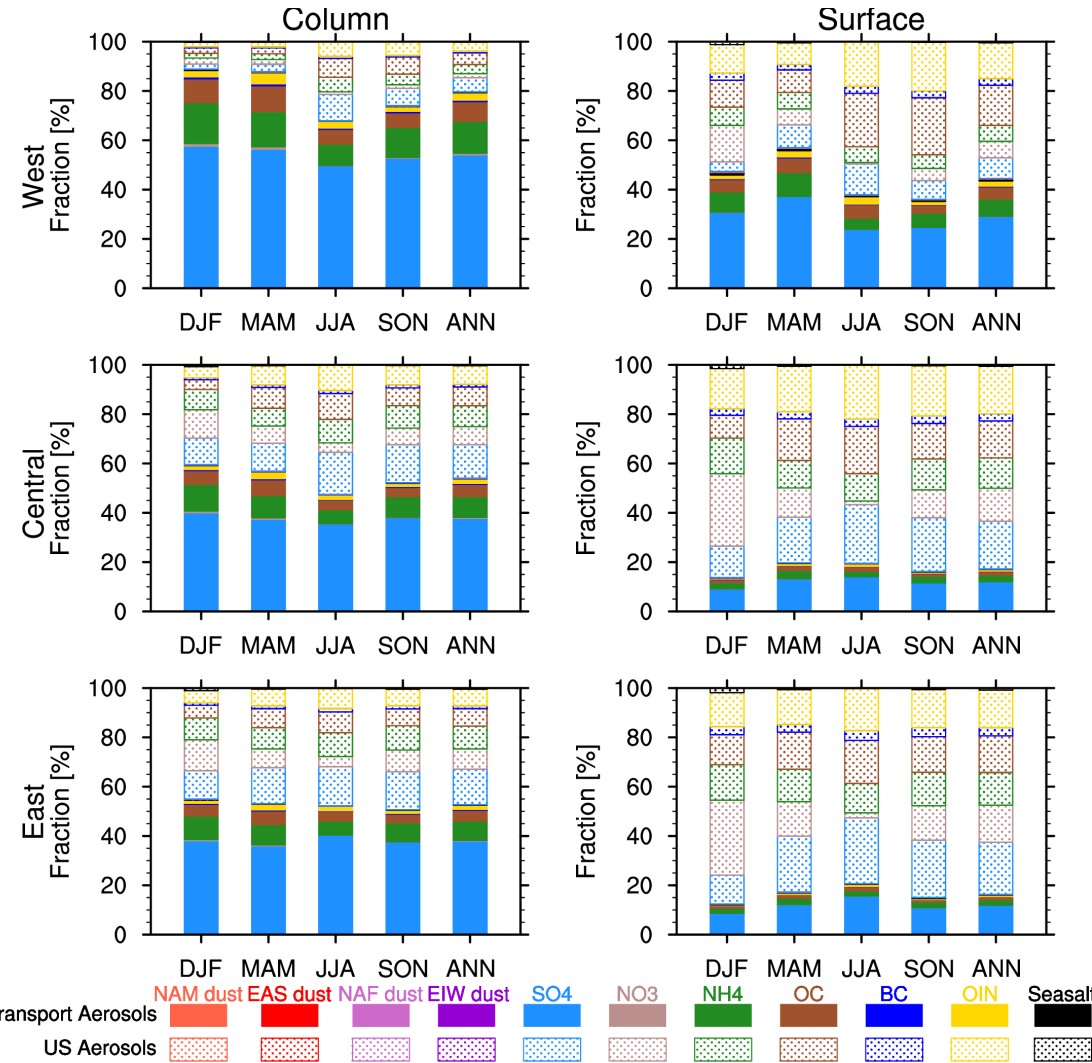

**Figure 14.** Same as in Fig. 13, but for aerosol number concentration.





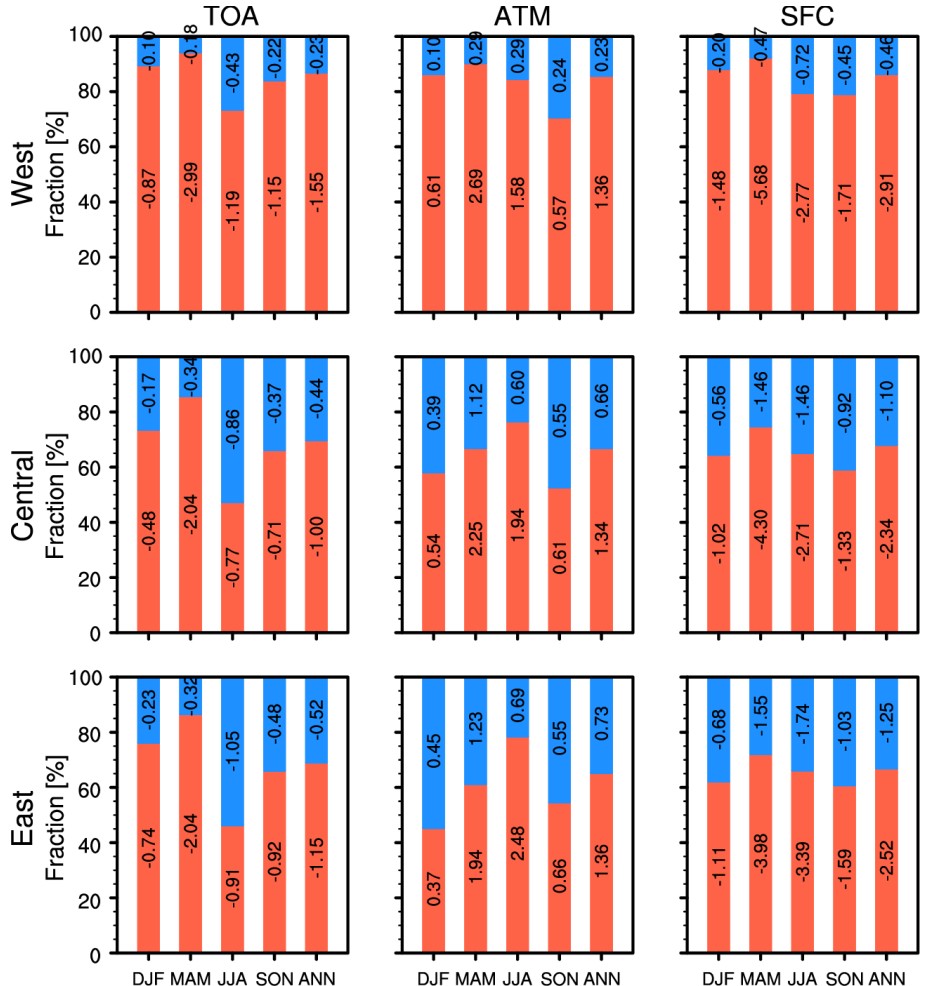

**Figure 15.** Seasonal and annual variation of radiative forcing fraction at the TOA (left), in the atmosphere (middle) and at

the surface (right) from trans-Pacific transport (red) and US (blue) aerosols in the western, central and eastern North America

for 2010-2014.