# Peer review of "Trans-Pacific transport and evolution of aerosols: Spatiotemporal characteristics and source contributions"

_Atmospheric Chemistry and Physics, 2019_

## Referee Comment (RC1) · Anonymous Referee #2 · 13 Jun 2019

Review on the "Trans-Pacific transport and evolution of aerosols: Spatiotemporal characteristics and source contributions" by Hu et al.

I went through my previous comments and I would say the authors addressed well for the majority of my previous comments. Again, I recommend this paper for the publication with minor revision. There are a few minor comments for further improvement.

Comments:

P4L21-P5L2: The current simulation was made in 1 degree resolution which is quite common in global models, for example several models participating AeroCom experiments. The WRF model can run with even higher resolution. What are the sensitivity to the different horizontal resolution? And please comment what are the advantage of your WRF model compared to global models.

P5L18-21: It needs to be more quantitative. Please add some statistics. What are bias, rmse, correlation, and/or std between model and obs?

P7L10-12: This sentence does not read well. Please restate this sentence.

P8L26-27: The sentence about dust is too brief. What region is considered here? Please add dust contribution below 4 km.

P9L15-16: Please provide more detailed explanation how the mass flux is calculated. Which level of wind is used? And how wind information is combined with mass? Which direction is considered in the calculation and why? Is the flux calculation conducted every time step or monthly or annually? Adding these additions should not be difficult.

P11L25-30: DRF ATM especially for warming by BC and dust needs comparison with other estimates. Please add citations and range.

Figure 11 and text: Again, BC contribution in Figure 11 is so dominant. Please compare with other estimates.

---

## Referee Comment (RC2) · Anonymous Referee #1 · 17 Jun 2019

This manuscript reports a quite comprehensive analysis of trans-Pacific transport and evolution of aerosol and its contribution to aerosol over North America based on 5-year quasi-global WRF-Chem simulations. The analysis includes both aerosol mass and number concentration, total and composition, surface and profile/column, direct radiative forcing and air quality implication. This study highlights the importance of trans-Pacific aerosol in determining aerosol column mass loading, number concentration, and direct radiative forcing, which is generally consistent with results of previous observational and modeling studies. This modeling study also provides more insights into composition of aerosol. It adds useful contribution to the discussion of long-range transport of aerosol and its climate and air quality impacts. I recommend the paper be

published in ACP after the following concerns(mostly minor) are adequately addressed.

1. Section 3.1.2: Many numbers appear in this section but often it is not clear what these numbers represent. For example, page 7 1st line: "The EAS dust decrease rapidly (7.60 ug/m2) during transport...." It is hard to understand what this 7.60 ug/m2 represents. Please check throughout this section (even the paper) to clarify what those numbers represent.

2. Section 3.2: this section calculates aerosol fluxes and compares with some observational estimate. Discussion about significant model-observation differences could be more thorough from both measurement and modeling perspective. One point I don't quite agree is that they attribute observation-model difference to different time periods and claimed that the "discrepancies may be due to the increasing pollution over East Asia under fast economic development". But many studies have shown that the pollution aerosol has been reducing since 2007/2008. It is also necessary to present and discuss the transport efficiency (aerosol flux arriving in North America to that leaving East Asia) and its variations with season and composition (e.g., dust vs pollution aerosol).

3. Sections 3.3 & 3.4.2 - Aerosol direct radiative forcing. We know that aerosol direct radiative forcing can take place in both clear and cloudy sky, in solar/shortwave and thermal infrared spectra. It is necessary to state clearly what you are estimating. We all know that the aerosol direct radiative forcing is determined by aerosol optical depth (AOD), single-scattering albedo (SSA), and phase function. But the paper never shows these quantities. I would suggest that they at least show AOD and SSA from transported aerosol vs North American aerosol. Then readers may understand why the transported aerosol causes much larger direct radiative forcing than the North American aerosol does.

4. Check throughout the paper and make sure that any acronyms are spelled out when they first appear.

5. page 5, line 29-30: "....due to the offset by the westward transport..." what do you mean here? Also do emissions change from season to season? does such the seasonal variation of emissions contribute?

6. page 7, line 24-25: "Over the Pacific Ocean...., while over the western Pacific...... " it is confusing.

7. page 8, line 18-20: Do you want to point to any figure (e.g., Figure 6) that shows "the nitrate is mainly concentrated in the low level ...." ?

8. Figure 1 caption: explain what PM10 represents

9. Figure 2: averaged over what longitudes?

10. Figure 3: can you overlay the percentage contribution of each component in the map, probably as isopleth?

11. Figure 4 caption: state clearly what are shown in left panels vs right panels.

12. Figure 10: can you explain why sulfate causes a warming effect in the atmosphere if sulfate is purely scattering aerosol as many studies have suggested?

---

## Author Comment (AC1) · 28 Jul 2019

We thank the two anonymous referees for their valuable comments and constructive suggestions on the manuscript. Below, we explain how the comments and suggestions are addressed and make note of the revision we made in the manuscript.

**Anonymous Referee #1**

*General comments:*

- *I went through my previous comments and I would say the authors addressed well for the majority of my previous comments. Again, I recommend this paper for the publication with minor revision. There are a few minor comments for further improvement.*

**Response:** We thank the reviewer for a detailed review. Both text and figures are revised as the reviewer suggested.

*minor comments:*

- *P4L21-P5L2: The current simulation was made in 1 degree resolution which is quite common in global models, for example several models participating AeroCom experiments. The WRF model can run with even higher resolution. What are the sensitivity to the different horizontal resolution? And please comment what are the advantage of your WRF model compared to global models.*

**Response:** Thank you so much for your suggestions. Generally, the performance of WRF-Chem model is sensitive to horizontal resolution. When increasing the horizontal resolution, the large geographical gradient over the complex topography can be well resolved (Zhao et al., 2013b; Hu et al., 2016), and the uncertainties in the estimation of cloud susceptibility to aerosols during long-range transport could be reduced (Ma et al., 2014). Also, increasing model resolution to reduce the sub-grid inhomogeneity and to better resolve aerosol plumes (Weigum et al., 2012) may reduce the model bias. So, we agree that the simulations with higher resolution will be better to capture the aerosol long-range transport. However, the WRF-Chem simulations configured at a global scale with higher resolution need much more computational cost, which is difficult for us at present. In this study, we used WRF-Chem model at 1 degree resolution, because this model has several advantages compared with global model (described in the following section). Also, this simulations at 1 degree resolution can provide reasonable inflow chemical boundaries to study regional air quality and climate with high-resolution nested regional modeling (e.g., Sarangi, et al., 2019; Shrivastava et al., 2019)

The quasi-global WRF-Chem model at 1 degree resolution can capture the characteristics of trans-Pacific aerosols. "Compared with global models, the USTC version of WRF-Chem used in this study 1) provides a relatively more accurate 8-bin approach to simulate the aerosol mass balance and radiative forcing, which can well simulate particle size distribution and thus aerosol lifetime during long-range transport (Zhao et al., 2013b); 2) includes complex aerosol processes and interactions between aerosol and radiation, clouds, and snow albedo, which can well resolve aerosol-cloud-precipitation interaction (Zhao et al., 2011; Zhao et al., 2012, Zhao et al., 2014; Hu et al., 2016); and 3) diagnoses radiative forcing of aerosol composition (Zhao et al., 2013a), which is not included in most global models that treat aerosols as internal mixing." (add in the P4L19-26)

Compared with global models, the USTC version of WRF-Chem used in this study 1) provides a relatively more accurate 8-bin approach to simulate the aerosol mass balance and radiative forcing, which can well simulate particle size distribution and thus aerosol lifetime during long-range transport (Zhao et al., 2013b); 2) includes complex aerosol processes and interactions between aerosol and radiation, clouds, and snow albedo, which can well resolve aerosol-cloud-precipitation interaction (Zhao et al., 2011; Zhao et al., 2012, Zhao et al., 2014; Hu et al., 2016); and 3) diagnoses radiative forcing of aerosol composition (Zhao et al., 2013a), which is not included in most global models that treat aerosols as internal mixing.

Hu, Z., Zhao, C., Huang, J., Leung, L. R., Qian, Y., Yu, H., Huang, L., and Kalashnikova, O. V.: Trans-Pacific transport and evolution of aerosols: evaluation of quasi-global WRF-Chem simulation with multiple observations, Geosci. Model Dev., 9, 1725–1746, https://doi.org/10.5194/gmd-9–1725–2016, 2016.

Shrivastava, M., Andreae, M., Artaxo, P., Barbosa, H. M. J., Berg, L. K., Brito, J., Ching, J., Easter, R.C., Fan, J., Fast, J. D., Feng, Z., Fuentes, J.D., Glasius, M., Goldstein, A. H., Alves, E. G., Gomes, H., Gu, D., Guenther, A., Jathar, S. H., Kim, S., Liu, Y., Lou, S., Martin, S. T., McNeill, V. F., Medeiros, A., de Sá, S. S., Shilling, J. E., Springston, S. R., Souza, R. A. F., Thornton, J. A., Isaacman-VanWertz, G., Yee, L. D., Ynoue, R., Zaveri, R. A., Zelenyuk, A., and Zhao, C.: Urban pollution

greatly enhances formation of natural aerosols over the Amazon rainforest, Nat. Commun., 10(1), https://doi.org/10.1038/s41467-019-08909-4.

Sarangi, C., Qian, Y., Rittger, K., Bormann, K. J., Liu, Y., Wang, H., Wan, H., Lin, G., and Painter, T. H.: Impact of light-absorbing particles on snow albedo darkening and associated radiative forcing over high-mountain Asia: high-resolution WRF-Chem modeling and new satellite observations, Atmos. Chem. Phys., 19, 7105-7128, https://doi.org/10.5194/acp-19-7105-2019, 2019.

Zhao, C., Liu, X., Ruby Leung, L., and Hagos, S.: Radiative impact of mineral dust on monsoon precipitation variability over West Africa, Atmos. Chem. Phys., 11, 1879–1893, doi:10.5194/acp–11–1879–2011, 2011.

Zhao, C., Liu, X., and Leung, L. R.: Impact of the Desert dust on the summer monsoon system over Southwestern North America, Atmos. Chem. Phys., 12, 3717–3731, doi:10.5194/acp–12-3717–2012, 2012.

Zhao, C., Leung, L. R., Easter, R., Hand, J., and Avise, J.: Characterization of speciated aerosol direct radiative forcing over California, J. Geophys. Res., 118, 2372–2388, doi:10.1029/2012JD018364, 2013a.

Zhao, C., Chen, S., Leung, L. R., Qian, Y., Kok, J. F., Zaveri, R. A., and Huang, J.: Uncertainty in modeling dust mass balance and radiative forcing from size parameterization, Atmos. Chem. Phys.,13, 10733–10753, doi:10.5194/acp–13–10733–2013, 2013b.

Zhao, C., Hu, Z., Qian, Y., Ruby Leung, L., Huang, J., Huang, M., Jin, J., Flanner, M. G., Zhang, R., Wang, H., Yan, H., Lu, Z., and Streets, D. G.: Simulating black carbon and dust and their radiative forcing in seasonal snow: a case study over North China with field campaign measurements, Atmos. Chem. Phys., 14, 11475–11491, doi:10.5194/acp–14–11475–2014, 2014.

- ***P5L18-21: It needs to be more quantitative. Please add some statistics. What are bias, rmse, correlation, and/or std between model and obs?***

**Response:** Thank you so much for your suggestions. Now we revise it as "Firstly, Hu et al. (2016) evaluated the simulations with various satellite retrievals and surface measurements, including AOD (aerosol optical depth) from MODIS, MISR (Multi-angle

Imaging SpectroRadiometer), OMI (Ozone Monitoring Instrument) and AERONET (AErosol RObotic NETwork), aerosol extinction coefficients from CALIPSO, and the surface mass concentration from IMPROVE (Interagency Monitoring of PROtected Visual Environments). The difference of annual mean AOD between simulations and MODIS (MISR) retrievals was about −0.01 (+0.01), 0 (0), and −0.01 (−0.01) over the western, central, and the eastern Pacific. Across the Pacific, the spatial correlation coefficients of AOD between simulations and satellite retrievals (e.g., MODIS and MISR) were ranged in 0.63~0.88 for the four seasons. Also, compared with AERONET measurements, about 90% of simulated AOD was within a factor of 2, and their monthly correlation coefficients were about 0.64~0.76. Additionly, the simulations could well capture the magnitude of aerosol surface mass concentrations over the western USA with a correlation coefficients of 0.75~0.83. Overall, these results shown that that the simulations well captured the spatial distribution and vertical profile characteristics of trans-Pacific transport aerosols."

- ***P7L10-12: This sentence does not read well. Please restate this sentence.***

**Response:** Thank you so much for your suggestions. Now we revise it as "In general, the trans-Pacific transported dust mass loading is about 1~2 times larger of the transported pollution in this region, and the transported dust at the surface is 2~4 times higher than that of pollution, which has been discussed by Chin et al. (2007)."

- ***P8L26-27: The sentence about dust is too brief. What region is considered here? Please add dust contribution below 4 km.***

**Response:** Thank you so much for your suggestions. Now we revise it as "Dust mass from three sources dominates the total aerosol mass concentrations above 4 km (about 60%) over the western, central, and eastern Pacific, while, below 4 km, the dust mass contributions are varying with a range of 5-58%."

- ***P9L15-16: Please provide more detailed explanation how the mass flux is calculated. Which level of wind is used? And how wind information is combined with mass? Which direction is considered in the calculation and why? Is the flux***

*calculation conducted every time step or monthly or annually? Adding these additions should not be difficult.*

**Response:** Thank you so much for your suggestions. We have added the calculation of the mass flux in the sections 2.3 (P5L21-29) as following text:

**2.3 Calculating the mass flux of aerosol component**

Firstly, the aerosol mass concentration (µg m$^{-3}$), east-west wind component U (m s$^{-1}$), and a segment with a length of L (m) that is a width of 10° in longitude, are used to calculate the aerosol mass flux (µg m$^{-1}$ s$^{-1}$) at each layer; then the mass flux multiplies the vertical height of each layer, and then is aggregated to columnar fluxes. Finally, the calculated 3-hourly fluxes are aggregated into a year. The detailed methodology is followed Yu et al. (2008):

$$F_p = \sum_{l=1}^{35} M_p(l)U(l)*L*H \qquad (1)$$

Where the $F_p$ is the columnar mass fluxes for aerosols (µg s$^{-1}$),the $M_p$ is aerosol mass concentration (µg m$^{-3}$), $U$ is east-west wind component (m s$^{-1}$), $L$ is the length with a width of 10° in longitude (m), $H$ is the vertical height of each layer (m), $l$ is model layer.

In this study, the east-west (zonal) wind is considered. The reason is that the aerosols are transported by the atmospheric circulation, in which the direction is east-west (zonal). Aslo, the aerosol outflowed and imported are mainly in meridional vertical section and this considered in the calculation is followed previous studies (e.g., Yu et al., 2008; 2012; Hu et al., 2019).

Hu, Z., Huang, J., Zhao, C., Bi, J., Jin, Q., Qian, Y., Leung, L. R., Feng, T., Chen, S., Ma, J.: Modeling the contributions of Northern Hemisphere dust sources to dust outflow from East Asia, Atmos. Environ., 202, 234-243, https://doi.org/10.1016/j.atmosenv.2019.01.022, 2019.

Yu, H. B., Remer, L. A., Chin, M., Bian, H. S., Kleidman, R. G., and Diehl, T.: A satellite-based assessment of transpacific transport of pollution aerosol, J. Geophys. Res., 113, D14S12, doi:10.1029/2007JD009349, 2008.

Yu, H., Remer, L. A., Chin, M., Bian, H., Tan, Q., Yuan, T., and Zhang, Y.: Aerosols from Overseas Rival Domestic Emissions over North America, Science, 337, 566–569, 2012.

- *P11L25-30: DRF ATM especially for warming by BC and dust needs comparison with other estimates. Please add citations and range.*

**Response:** Thank you so much for your suggestions. In the atmosphere, the BC and dust have ability to absorb and/or scatter sunlight (Bond et al., 2013; Zhao et al., 2013; Chen et al., 2013), and then warm the atmosphere with a positive direct radiative forcing. In order to supporting our results, the BC radiative forcing in the atmosphere is compared with the results from CAM5 simulations (Fig. R1, Jones et al. 2007; Bond et al., 2013) and the dust radiative forcing in the atmosphere is compared with the results from ECHAM/MESSy atmospheric chemistry climate model simulations (Fig. R2, Klingmüller et al., 2019). It shows that the spatial distribution of both results is consist and the magnitude of BC and dust radiative forcing in the atmosphere is closer. The BC radiative forcing in the atmosphere from our model simulations is ranged in $+1 \sim +8$ W m$^{-2}$, which is well consist with the results of the Hadley Centre climate model (HadGEM1) simulations with a range of $+1 \sim +7.5$ W m$^{-2}$. For the dust radiative forcing in the atmosphere, the range is about $+0.1 \sim +0.4$ W m$^{-2}$ in our results over the western, central and eastern Pacific, which is well consist with the results of ECHAM/MESSy atmospheric chemistry climate model simulations with a range of $+0.2 \sim +0.4$ W m$^{-2}$.

   Also, we revise the sentences in *P12L25-31* as: "In the ATM, aerosol compositions lead to a warming effect, with black carbon inducing the largest warming of about $+8$ W m$^{-2}$, and the dust and sulfate direct radiative forcing is surprisingly much small even though they have larger mass. This can be attributed to the strongest absorbing property of black carbon (Bond et al., 2013). Dust produces a warming effect with a maximum value of about $+2.0$ W m$^{-2}$ and a domain average of $+0.13$ W m$^{-2}$ in the ATM. Over the western, central and eastern Pacific, the dust and BC radiative forcing is ranged in $+0.1 \sim +0.4$ W m$^{-2}$ and $+1 \sim +8$ W m$^{-2}$, respectively, which is well consist with the results of global model simulations with a range of $+0.2 \sim +0.4$ W m$^{-2}$ (Klingmüller et al., 2019) and $+1 \sim +7.5$ W m$^{-2}$ (Jones et al. 2007; Bond et al., 2013)."

[Figure]

**Figure R1.** Direct radiative forcing due to BC aerosols (W m$^{-2}$) from the ECHAM/MESSy atmospheric chemistry climate model. From Bond et al. (2013) and Jones et al. (2007).

[Figure]

**Figure R2.** Direct radiative forcing due to dust aerosols (W m$^{-2}$) from the Hadley Centre climate model (HadGEM1). From Klingmüller et al. (2019).

Bond, T. C., Doherty, S. J., Fahey, D. W., Forster, P. M., Berntsen, T., DeAngelo, B. J., Flanner, M. G., Chan, S., Kärcher, B., Koch, D., Kinne, D., Kondo, Y., Quinn, P. K.,Sarofim, M. C., Schultz, M. G., Schulz, M., Venkataraman, C., Zhang, H., Zhang, S., Bellouin, N., Guttikunda, S. K., Hopke, P. K., Jacobo, M. Z., Kaiser, J. W., Klimont, Z., Lohmann, U., Schwarz, J. P., Shindell, D., Storelvmo, T., Warren, S. G., and Zender C. S.: Bounding the role of black carbon in the climate

system: A scientific assessment, J. Geophys. Res. Atmos., 118,5380–5552, doi:10.1002/jgrd.50171, 2013.

Jones, A., Haywood, J. M., and Boucher, O.: Aerosol forcing, climate response and climate sensitivity in the Hadley Centre climate model, J. Geophys. Res., 112(D20), 211, doi:10.1029/2007JD008688, 2007.

Klingmüller, K., Lelieveld, J., Karydis, V. A., and Stenchikov, G. L.: Direct radiative effect of dust–pollution interactions, Atmos. Chem. Phys., 19, 7397-7408, https://doi.org/10.5194/acp-19-7397-2019, 2019.

- *Figure 11 and text: Again, BC contribution in Figure 11 is so dominant. Please compare with other estimates.*

**Response:** Thank you so much for your suggestions. Over the eastern Pacific, previous studied (e.g., Chung et al. 2005, 2010; Wu et al., 2008; Zhuang et al., 2011; Gao et al., 2014) had calculated the BC radiative forcing, and the range was about –5.3 ~ –2.9 W m$^{-2}$ at the surface, +4.6 ~ +7.6 W m$^{-2}$ in the atmosphere, and +1.0 ~ +2.3 W m$^{-2}$ at the TOA, respectively. In our results, the BC radiative forcing is about –2.8 W m$^{-2}$ at the surface, +4.2 W m$^{-2}$ in the atmosphere, and +1.5 W m$^{-2}$ at the TOA, respectively. We can see that our results is consistent with previous studies. Also, Gao et al (2014) and Chung et al. (2010) had shown that the BC dominated the contribution of radiative forcing, which was consistent with our results. Over the western Pacific, we compare the radiative forcing of BC over this region with that over California shown in Zhao et al. (2013a) (Fig. R3). The BC radiative forcing at TOA is positive value and is larger than other aerosols. In the atmosphere, the value of BC radiative forcing dominates the total radiative forcing. Further, we can see that the value of BC radiative forcing from Zhao et al. (2013a) is closed with our results over the western Pacific. Over the central Pacific, the spatial distribution of BC radiative forcing in the atmosphere is consistent with previous studies (Fig. R1, Jones et al. 2007; Bond et al., 2013). However, we cannot find the regional mean BC radiative forcing in previous studies. Overall, we think the aerosol compositional radiative forcing in our study is relatively reasonable.

[Figure]

**Figure R3.** Seasonal variations of aerosol direct radiative forcing and its contributions from sulfate, OM, EC, dust, and other species at the TOA, in the atmosphere, and at the surface from the WRF-Chem simulations with anthropogenic EC emission doubled. Other species include nitrate, ammonium, sea salt, and unspeciated PM2.5. Black and purple bars represent the total forcing from the simulation and the sum of the diagnosed individual forcings. From Zhao et al. (2013a).

Chung, C. E., Ramanathan, V., Carmichael, G., Kulkarni, S., Tang, Y., Adhikary, B., Leung, L.R., Qian, Y.: Anthropogenic aerosol radiative forcing in Asia derived from regional models with atmospheric and aerosol data assimilation. Atmos. Chem. Phys., 10, 6007e6024, 2010.

Gao, Y., Zhao, C., Liu, X., Zhang, M., and Leung, L. R.: Regional modeling of aerosol and its radiative forcing over East Asia using WRF-Chem, Atmos. Environ., 92, 250–266, 2014.

Zhao, C., Leung, L. R., Easter, R., Hand, J., and Avise, J.: Characterization of speciated aerosol direct radiative forcing over California, J. Geophys. Res., 118, 2372–2388, doi:10.1029/2012JD018364, 2013a.

**Anonymous Referee #2**

*General comments:*

- *This manuscript reports a quite comprehensive analysis of trans-Pacific transport and evolution of aerosol and its contribution to aerosol over North America based on 5-year quasi-global WRF-Chem simulations. The analysis includes both aerosol mass and number concentration, total and composition, surface and profile/column, direct radiative forcing and air quality implication. This study highlights the importance of trans-Pacific aerosol in determining aerosol column mass loading, number concentration, and direct radiative forcing, which is generally consistent with results of previous observational and modeling studies. This modeling study also provides more insights into composition of aerosol. It adds useful contribution to the discussion of long-range transport of aerosol and its climate and air quality impacts. I recommend the paper be published in ACP after the following concerns(mostly minor) are adequately addressed.*

**Response:** We thank the reviewer for a detailed review. Both text and figures are revised as the reviewer suggested.

*Specific comments:*

- *1. Section 3.1.2: Many numbers appear in this section but often it is not clear what these numbers represent. For example, page 7 1st line: "The EAS dust decrease rapidly (7.60 ug/m2) during transport...." It is hard to understand what this 7.60 ug/m2 represents. Please check throughout this section (even the paper) to clarify what those numbers represent.*

**Response:** Thank you so much for your suggestions. We have revised these numbers and now their meanings are more clearly represented throughout the paper. For details, please see Section 3.1.2 in the revised manuscript.

- *2. Section 3.2: this section calculates aerosol fluxes and compares with some observational estimate. Discussion about significant model-observation differences could be more thorough from both measurement and modeling perspective. One point I don't quite agree is that they attribute observation-model difference to*

*different time periods and claimed that the "discrepancies may be due to the increasing pollution over east Asia under fast economic development". But many studies have shown that the pollution aerosol has been reducing since 2007/2008. It is also necessary to present and discuss the transport efficiency (aerosol flux arriving in North America to that leaving East Asia) and its variations with season and composition (e.g., dust vs pollution aerosol).*

**Response:** Thank you so much for your suggestions. We changed the explaination as "Because this comparison is complicated by differences in the time period, the discrepancies may be due to the larger variabilities of aerosols over East Asia.". Also, we added the discussion of aerosol transport efficiency (aerosol flux arriving into North America *vs* that leaving East Asia) and its variations with season in the manuscript (in the *page11 line32-34* and *page12 line1*) as "Additionally, the aerosol transport efficiency (aerosol mass flux arriving into North America *vs* that leaving East Asia) is about 34% at the annual timescale with a peak value of 40% in JJA, in which the transport efficiency of dust is in 41% in JJA, while the transport efficientcy of the pollution aerosol (included sulfate, nitrate, organic matter, black carbon and ammonium) is 43% in MAM  (Figure S1)."

[Figure]

**Figure S1.** The transport efficiency of aerosol mass flux arriving into North America vs that leaving East Asia in seasonal and annual from total aerosol (included dust, sulfate, nitrate, organic matter, black carbon and ammonium), dust, and pollution aerosol (included sulfate, nitrate, organic matter, black carbon and ammonium).

- *3. Sections 3.3 & 3.4.2 - Aerosol direct radiative forcing. We know that aerosol direct radiative forcing can take place in both clear and cloudy sky, in solar/shortwave and thermal infrared spectra. It is necessary to state clearly what you are estimating. We all know that the aerosol direct radiative forcing is determined by aerosol optical depth (AOD), single-scattering albedo (SSA), and phase function. But the paper never shows these quantities. I would suggest that they at least show AOD and SSA from transported aerosol vs North American aerosol. Then readers may understand why the transported aerosol causes much larger direct radiative forcing than the North American aerosol does.*

**Response:** Thank you so much for your suggestions. In the manuscript, we have stated the aerosol direct radiative forcing is "under all-sky conditions for the net (shortwave + longwave) radiation" in *page12 line17-18*. For the quantities of simulations used in this study, we have evaluated and discussed in our previous study (Hu et al., 2016). These results shown that the simulations could well capture the spatial distribution of aerosol optical depth (AOD) and its seasonal variation compared with satellite and AERONET retrievals. Also, the model reasonably simulated the variability of absorbing AOD (AAOD) and aerosol size distributions, which was consist with the retrievals from OMI and MODIS. Compared with the retrievals from CALIPSO, the vertical profile of aerosol extinction was simulated by the model. So, we think the model can well simulate the distribution of aerosols and these optical characteristics.

Additionally, we show the spatial distribution of AOD, AAOD and single-scattering albedo (SSA) from transported aerosol and North American aerosol (Figure S2). "Clearly, the AOD and AAOD over the East Asia, North Pacific and the west coast of North America are dominated by the transported aerosol, while the AOD and AAOD over the eastern North America are dominated North American aerosol. Also, the SSA from transported aerosol are larger than that from North American aerosol over the North America (excepted the northeastern region of North America). Therefore, the transported aerosol causes much larger direct radiative forcing." This discussion has been added in *page12 line12-16*.

[Figure]

**Figure S2.** Spatial distribution of aerosol optical depth, absorbing aerosol optical depth, and single scatter albedo from the transported aerosol and North American aerosol. The WRF-Chem simulation is averaged for 2010-2014.

- *4. Check throughout the paper and make sure that any acronyms are spelled out when they first appear.*

**Response:** We have checked and make sure that any acronyms are spelled out at their first appearance.

- *5. page 5, line 29-30: "....due to the offset by the westward transport..." what do you mean here? Also do emissions change from season to season? does such the seasonal variation of emissions contribute?*

**Response:** Thank you so much for your suggestions. In this study, the emissions is changed from season to season, so we revise it as "The peak trans-Pacific aerosol mass concentrations occur in spring (MAM) due to the strongest mid-latitude westerlies and active extratropical cyclones (particularly south of 30° N) (Yu et al., 2012), and more emissions in this season (Yu et al., 2008), while the minimum occurs in summer (JJA) because of the greatest aerosol removal induced by summer monsoon precipitation (Holzer et al., 2005; Yu et al., 2008, 2013)."

- *6. page 7, line 24-25: "Over the Pacific Ocean...., while over the western Pacific...... " it is confusing.*

**Response:** Thank you so much for your suggestions. we revise it as "Below 1 km, the seasalt is the dominant aerosol over the Pacific Ocean, and the dust and pollution aerosols outflow from East Asia make significant contributions over the western Pacific."

- *7. page 8, line 18-20: Do you want to point to any figure (e.g., Figure 6) that shows "the nitrate is mainly concentrated in the low level ...." ?*

**Response:** Thank you so much for your suggestions. We revise it as "During the transport over the central and eastern Pacific, the contribution of sulfate becomes more than 60%. Relatively, the nitrate decreases rapidly to less than 4% during the trans-Pacific transport, which is likely due to the fact that nitrate particles are mainly concentrated in the low level over the western Pacific (shown in Fig. 6) and can be removed easily during the transport."

- *8. Figure 1 caption: explain what PM10 represents*

**Response:** Thank you so much for your suggestions. we revise it as "**Figure 1.** Spatial distribution of seasonal mean aerosol column mass loading from WRF-Chem averaged for 2010-2014. Three regions are denoted by the black boxes: the western Pacific (20° N – 50° N and 120° E – 140° E), the central Pacific (20° N – 50°N and 140° E – 140° W), and the eastern Pacific (20° N – 50° N and 140° W –120° W) for analysis. PM10 is including the mineral dust, sulfate (SO4), nitrate (NO3), ammonium (NH4), organic matter (OM), black carbon (BC), other inorganic matter (OIN), and sea-salt."

- *9. Figure 2: averaged over what longitudes?*

**Response:** Thank you so much for your suggestions. We have add the range of longitudes in the caption of Figure 2 as "**Figure 2** Vertical cross-section of zonal mean aerosol mass concentration averaged from 120° E to 120 ° W for each season from the WRF-Chem simulation averaged for 2010-2014."

- *10. Figure 3: can you overlay the percentage contribution of each component in the map, probably as isopleth?*

**Response:** Thank you so much for your suggestions. We have overlaid the percentage contribution of each component in the map as isopleth in Figure 3.

[Figure]

**Figure 3.** Spatial distribution of aerosol composition column mass loading from WRF-Chem averaged for 2010-2014. The trans-Pacific transport aerosol mass spatial distribution is denoted by PM10 and the dust from North America, East Asia, North Africa, elsewhere in the world are denoted by NAM dust, EAS dust, NAF dust, EIW dust. Contour lines (blue solid line) denote the percentage contribution (%) of the corresponding aerosol composition to total aerosol column mass.

- *11. Figure 4 caption: state clearly what are shown in left panels vs right panels.*

**Response:** Thank you so much for your suggestions. We have revise it as "**Figure 4.** Vertical cross-section of zonal mean aerosol mass concentration (left panels: a, b, c) and vertical distributions of mean aerosol mass (blue solid line; upper $x$ axis) and the composition fractions (colored shade-contour; lower $x$ axis) (right panels: d, e, f) from the WRF-Chem simulation averaged for 2010-2014 over the three regions shown in Fig. 1."

- *12. Figure 10: can you explain why sulfate causes a warming effect in the atmosphere if sulfate is purely scattering aerosol as many studies have suggested?*

**Response:** Thank you so much for your suggestions. Generally, the sulfate is a type of nonabsorbing aerosols, which can only scattering the solar-light and cause cooling effect in the atmosphere. In this study, the model consider internal mixing of nonabsorbing aerosols (e.g., sulfate) with BC and dust, and the warming effect of sulfate is the enhancement of BC and dust warming by all sulfate aerosols through internal mixing (Zhao et al. 2013a). Also, the radiative forcing of sulfate over the western Pacific from our simulation is consist with the results from Zhao et al. (2013a) over California.

Zhao, C., Leung, L. R., Easter, R., Hand, J., and Avise, J.: Characterization of speciated aerosol direct radiative forcing over California, J. Geophys. Res., 118, 2372–2388, doi:10.1029/2012JD018364, 2013a.